# Organization of olfactory centres in the malaria mosquito *Anopheles gambiae*

Olena Riabinina[1,†], Darya Task[1], Elizabeth Marr[1], Chun-Chieh Lin[1], Robert Alford[2], David A. O'Brochta[2] & Christopher J. Potter[1]

Mosquitoes are vectors for multiple infectious human diseases and use a variety of sensory cues (olfactory, temperature, humidity and visual) to locate a human host. A comprehensive understanding of the circuitry underlying sensory signalling in the mosquito brain is lacking. Here we used the Q-system of binary gene expression to develop transgenic lines of *Anopheles gambiae* in which olfactory receptor neurons expressing the odorant receptor co-receptor (*Orco*) gene are labelled with GFP. These neurons project from the antennae and maxillary palps to the antennal lobe (AL) and from the labella on the proboscis to the suboesophageal zone (SEZ), suggesting integration of olfactory and gustatory signals occurs in this brain region. We present detailed anatomical maps of olfactory innervations in the AL and the SEZ, identifying glomeruli that may respond to human body odours or carbon dioxide. Our results pave the way for anatomical and functional neurogenetic studies of sensory processing in mosquitoes.

[1] The Solomon H. Snyder Department of Neuroscience, Center for Sensory Biology, Johns Hopkins University School of Medicine, 855 North Wolfe Street, 434 Rangos Building, Baltimore, Maryland 21205, USA. [2] University of Maryland College Park, 9600 Gudelsky Drive, Rockville, Maryland 20850, USA. † Present address: MRC Clinical Sciences Centre, Imperial College London, Du Cane Road, London W12 0NN, UK. Correspondence and requests for materials should be addressed to C.J.P. (email: cpotter@jhmi.edu).

Anopheles gambiae is the major insect vector of *Plasmodium falciparum*, the parasite that leads to malaria which is responsible for ∼450,000 deaths every year[1]. The control of these insects is therefore an important human health issue. Currently, malaria control strategies focus on eliminating stages of parasite transmission (such as *Plasmodium* infection of mosquitoes), or to limit the size of mosquito populations[2]. However, relatively few strategies focus on interfering with the human host-seeking behavior of mosquitoes. Female *A. gambiae* mosquitoes are strongly anthropophilic and prefer to bloodfeed on humans over other animals. They have also been shown to use olfactory cues to locate humans[3,4] and are likely to use olfactory cues to distinguish humans from other animals[5–7]. Therefore, strategies that target the mosquito's olfactory system might provide effective means to control host seeking, including the development of improved trap baits and insect repellants. However, relatively little is known about the mechanisms of olfactory processing in *A. gambiae*.

Knowledge about insect olfactory systems primarily stems from studies of the vinegar fly *Drosophila melanogaster* (for example, see ref. 8). Over the last 25 years, the molecular and cellular basis of *Drosophila* olfaction has been well characterized (for example, see refs 9–12). Numerous studies have elucidated the identity of *Drosophila* olfactory receptors and the odours that activate them, the identity and structure of the sensory neurons that express these odorant receptors (ORs), how these olfactory receptor neurons innervate the brain, all leading to an in-depth understanding of how odorant information is received, integrated and processed by the *Drosophila* peripheral and central nervous systems. Studies of *Drosophila* olfaction are facilitated by a rich repertoire of genetic tools that are not currently available in mosquitoes. The development of similar neurogenetic methods in the mosquito system would greatly advance investigation of sensory processing in this insect.

In the absence of neurogenetic methods, studies in *A. gambiae* have focused on a detailed characterization of the peripheral olfactory system. Chemosensory tissues in malaria mosquitoes are covered in specialized sensory hairs called sensilla that typically contain two to three olfactory receptor neurons[13–15]. Similar to the peripheral olfactory system in *Drosophila*, 3 classes of chemosensory receptors are found in the *A. gambiae* genome: 79 ORs[12,16,17], 46 ionotropic receptors (IRs)[18,19] and 60 gustatory receptors (GRs)[20–23]. *In situ* hybridization and single sensillum electrophysiological recordings in *Drosophila*, mosquitoes and other insects show that neurons expressing ORs and IRs respond to a variety of volatile odorants including many found in human body odours[11,18,24–26], whereas GRs respond to a number of stimuli including tastants, pheromones, warmth and the volatile gas carbon dioxide[22,27].

ORs in all insects require an obligatory OR co-receptor (Orco) for dendritic trafficking of ORs and odorant-induced signalling[28,29]. Thus, all functional insect OR-expressing neurons also express the *Orco* gene. OR-expressing neurons have been detected by fluorescence *in situ* hybridization studies of *Orco* in the antennae and maxillary palps during the fourth instar larvae developmental stage of *A. gambiae*[30]. In the adult *A. gambiae*, Orco- and OR-expressing neurons have been found in three olfactory organs. Prominent Orco expression was found in the trichoid sensilla on the 2nd to 13th antennal segments in females, and on the distal club segments in males[17]. Orco expression has also been found in the capitate peg sensilla on the second to fourth segments of the maxillary palps in females and in the labella of the proboscis[17,22]. The maxillary palp capitate pegs in *A. aegypti*[31] and *A. gambiae*[22] also contain neurons that express a $CO_2$-sensing GR receptor complex. Orco-expressing neurons comprise ∼70% of all olfactory neurons in *Drosophila*, suggesting *Orco* is likely to be expressed in the majority of olfactory neurons in *A. gambiae* and thus is an attractive candidate for neurogenetic targeting in this species.

Binary expression systems, such as GAL4/*UAS*[32,33] and QF/*QUAS* (Q-system[34,35]), are useful and versatile tools for targeting distinct neuronal subpopulations in *Drosophila*[36]. The GAL4/*UAS* system has been used previously in *A. aegypti*[37], *A. gambiae*[38] and *Anopheles stephensi*[39] to target tissues such as the midgut that are involved in parasite infection in the mosquito. To enable robust *in vivo* activity, the activation domain of GAL4 was replaced with the activation domain of the Herpes simplex VP16, to achieve satisfactory expression of a *UAS-YFP* reporter[38] in *A. gambiae*. A second generation of the QF transcriptional activator, QF2, drives strong expression of reporters without compromising cellular functions or the overall health of transgenic animals[35]. QF2 is also likely to be a more potent *in vivo* transcriptional activator than GAL4 (refs 34,35).

Here we used the Q-system to target Orco-expressing olfactory neurons in *A. gambiae*. We characterized a sexually dimorphic expression pattern of Orco-expressing (Orco+) neurons in both larvae and adult mosquitoes. This transgenic labelling of olfactory neurons allowed visualization and identification of Orco+ axonal targets in the brain. Surprisingly, we found that the brain of *A. gambiae* contains two regions innervated by Orco+ olfactory neurons. Antennal and maxillary palp neurons target the glomeruli of the antennal lobe, but Orco+ olfactory neurons in the proboscis labella innervate the suboesophageal zone (SEZ) in the brain, a region likely to be involved in gustatory processing. We generated the first map of *A. gambiae* antennal lobe (AL) glomeruli based on olfactory receptor expression and tissue of origin. The AL contains approximately equal numbers of glomeruli innervated by Orco+ and Orco-negative (Orco−) neurons. We hypothesize that the majority of Orco− glomeruli in the AL probably receive innervation from antennal olfactory neurons expressing chemosensory IR genes. We also identified the Orco− AL glomeruli probably targeted by the GR-expressing (carbon dioxide-sensing) neurons from the maxillary palp. We further generated a map of the eight glomerular structures in the SEZ formed by Orco+ olfactory neurons.

These studies introduce a neurogenetic approach for investigating olfactory processing in the mosquito and suggest that the SEZ of the brain may directly integrate olfactory and gustatory cues from the proboscis during bloodfeeding.

## Results

**Introduction of neurogenetic tools into *A. gambiae*.** The introduction of the binary Q-system into *A. gambiae* required the generation of two different transgenic animal lines: a *promoter-QF2* driver line and a *QUAS-geneX* effector line. The QF2 transcriptional activator binds and activates the exogenous transgenic effector gene (*QUAS-geneX*). Only when the two components are brought together in the same animal by genetic crosses is expression of the effector gene induced in the same pattern as the transcriptional activator. To genetically target olfactory receptor neurons, we used the presumptive enhancer and promoter regions of the *Orco* (AGAP002560, VectorBase v.AgamP4) gene to generate an *Orco-QF2* driver construct. We PCR amplified a 9,312 bp sequence immediately upstream of the *Orco* ATG translation start codon and cloned it 5′ to QF2 in a piggyBac vector containing the synthetic *3xP3* eye promoter driving a DsRed fluorescent marker (see Supplementary Fig. 1 and Methods for details). The 9,312 bp putative promoter/enhancer region was the largest PCR product 5′ to the *Orco* ATG start codon we could obtain and it is possible that the

actual genomic region needed for Orco expression is located within a smaller portion of this sequence.

To facilitate the study of neuronal anatomy, we generated a membrane-targeted green fluorescent protein (GFP) reporter (*QUAS-mCD8:GFP*), which has been shown to strongly label all neuronal processes in *Drosophila*[40]. This allowed the axonal targets of the olfactory receptor neurons to be easily identified—a task not possible using existing anti-Orco antibodies. To generate the *QUAS-mCD8:GFP* reporter construct, we cloned 15 copies of the *QUAS* sequence[34] followed by the *Drosophila hsp70* minimal promoter into a piggyBac vector marked by a *3xP3-ECFP* marker. Immediately downstream of the *hsp70* promoter, we cloned a gene containing the mouse CD8 single-pass membrane protein fused to GFP (see Supplementary Fig. 1 and Methods for details). Increasing the number of transcription factor-binding sites in reporter constructs increases expression levels[41] and we reasoned that using *15x QUAS* instead of the standard *5x QUAS*[34] would lead to increased expression levels of GFP. This approach may prove particularly useful in reporting the activity of weak promoter lines, including olfactory receptor promoters[9,10,29].

We generated three *Orco-QF2* transgenic lines and two *QUAS-mCD8:GFP* transgenic lines. One *Orco-QF2* transgenic line drove expression in only a subset of olfactory neurons, likely to be due to genomic silencing of gene expression[42,43], and was not characterized further. The other two *Orco-QF2* lines produced highly similar expression patterns. Both *QUAS-mCD8:GFP* transgenic lines drove similarly robust QF2-dependent reporter expression. For the work presented here, a single *Orco-QF2* and a single *QUAS-mCD8:GFP* line were used. It should be noted that we did observe weak, uninduced expression from both *QUAS* reporters in a small subset of mushroom body and optic lobe neurons in adult brains, as well as weak expression in three cells in the larval maxillary palps (Supplementary Fig. 2). The nonspecific reporter expression did not interfere with the analyses of Orco + olfactory neurons.

**Transgenic expression pattern in larvae and adults**. We examined the expression pattern of the *Orco-QF2* driver line by crossing it to the *QUAS-mCD8:GFP* reporter. Transgenically driven GFP expression was robust and allowed direct visualization of GFP-labelled cell bodies and sensilla in larvae (Supplementary Fig. 3) and in adult mosquitoes without immunostaining (Fig. 1a,b and Supplementary Fig. 4). GFP fluorescence was evident in the antennae and maxillary palps of third and fourth instar larvae (Supplementary Fig. 3), and in the antennae, maxillary palps and proboscis of adult animals (Fig. 1). Larval antennal (female: median = 10 cells per antenna, interquartile range (IQR) = 2 cells per antenna, $n = 20$; male: median = 11 cells per antenna, IQR = 1.75 cells per antenna, $n = 14$) and maxillary palp (both female and male: median = 5 cells per palp, IQR = 1 cell per palp, $n = 20$) expression was similar between males and females (Supplementary Figs 3 and 5, and Supplementary Table 1). Expression patterns in the adult antennae and maxillary palps (Fig. 1c,d) were sexually dimorphic, consistent with previous reports[17,44]. Approximately 500–600 cells were labelled in the female antenna, with GFP + cells and trichoid sensilla densely distributed in the 12 distal antennal segments (Fig. 1a,c). In contrast, ~230 GFP + cells, located predominantly in the distal 2 segments, were detectable in the male antenna (Fig. 1b,d). Previous studies report that 95% of trichoid sensilla contain two olfactory neurons[13]. When combined with trichoid sensilla counts (~630 trichoid sensilla in females and ~225 in males)[15,45,46], this predicts ~1,250 olfactory neurons in female antennae and ~550 olfactory neurons in male antennae. Interestingly, electron microscopy

studies of *A. stephensi* antennae report five different types of trichoid sensilla based on physical characteristics[15,46]. Given the numbers of Orco + neurons identified here, this suggests that many types of trichoid sensilla may not be innervated by Orco + neurons, but may be innervated by other classes of olfactory neurons. It could also be possible that some or all trichoid sensilla contain one Orco + and one Orco − olfactory neuron, although this arrangement is rarely observed in *Drosophila*.

The maxillary palps in *A. gambiae* have ~67 capitate pegs in females and ~14 pegs in males[15]. We observed 120 GFP + cells in female maxillary palp capitate pegs (Fig. 1c,d) distributed along segments 2–4 of the palp (Fig. 1a,c) and 40 GFP + cells in the male maxillary palp sparsely distributed in the distal club (Fig. 1b,d). These numbers suggest that approximately two cells are labelled per capitate peg, in agreement with previous reports[22] that two Orco-dependent ORs (OR8 and OR28) are expressed in the maxillary palps. Expression in the proboscis shows a similar pattern in males and females, with ~45 GFP + cells located in the labella of the proboscis (Fig. 1c,d).

There were no GFP-labelled cells besides the antennae and maxillary palps in third to fourth instar larvae. In adults, we focused our investigation only on external tissues for *Orco-QF2, QUAS-mCD8GFP* expression, which we found was limited to the expression in the olfactory organs detailed above. It remains to be determined whether other tissues and cell types, such as sperm[47], are also labelled by *Orco-QF2*.

**Validation of expression**. To verify that the *Orco-QF2* driver accurately targets Orco-expressing neurons, we immunostained peripheral appendages of *Orco-QF2, QUAS-CD8:GFP* animals with antibodies directed against DmOrco[29] and CD8 (Fig. 2 and Supplementary Fig. 6), comparing immunofluorescence signals in these two channels with GFP fluorescence. Whole-mount antibody staining of olfactory tissues can be challenging due to limited access to neurons within cuticular tissues. Anti-CD8 acted as a control for the efficiency of antibody labelling of CD8:GFP protein and antibody staining was often less effective than labelling observed from native GFP fluorescence. In larval antennae and maxillary palps (Supplementary Fig. 6) 91–100% of Orco + cells were labelled by the anti-CD8 antibody, whereas 94–100% of CD8 + cells were also labelled by the anti-DmOrco antibody (Supplementary Table 2). Similarly, in the adult antennae, maxillary palps and labella, the overlap between anti-Orco and anti-CD8 staining included 100% of Orco + cells and 84–97% of CD8 + cells (Fig. 2 and Supplementary Table 3). We also performed sections of adult antennae and immunostained for anti-DmOrco (Supplementary Fig. 7). We found 100% of anti-DmOrco + sensilla were co-labelled by GFP (201 sensilla in 19 samples).

These results indicate that the *Orco-QF2* line robustly labels cells that are detectable with anti-DmOrco and also a small number of cells not detectable by anti-DmOrco. It is possible that these extra cells may represent Orco − olfactory neurons. Alternatively, as binary expression systems allow for amplification of weak promoter signals, it is plausible that the *QUAS-mCD8:GFP* reporter labels neurons that express low levels of Orco protein and are not detectable by the anti-DmOrco staining due to the background of this antibody when used in *Anopheles* tissues.

**Orco + neuronal targets in the adult mosquito brain**. Using our neurogenetic approach we find that olfactory receptor neurons labelled by the *Orco-QF2* driver line send their axons to two distinct areas of the brain: the AL and the SEZ (Fig. 3a). The AL is larger and has denser Orco + neuron innervation in

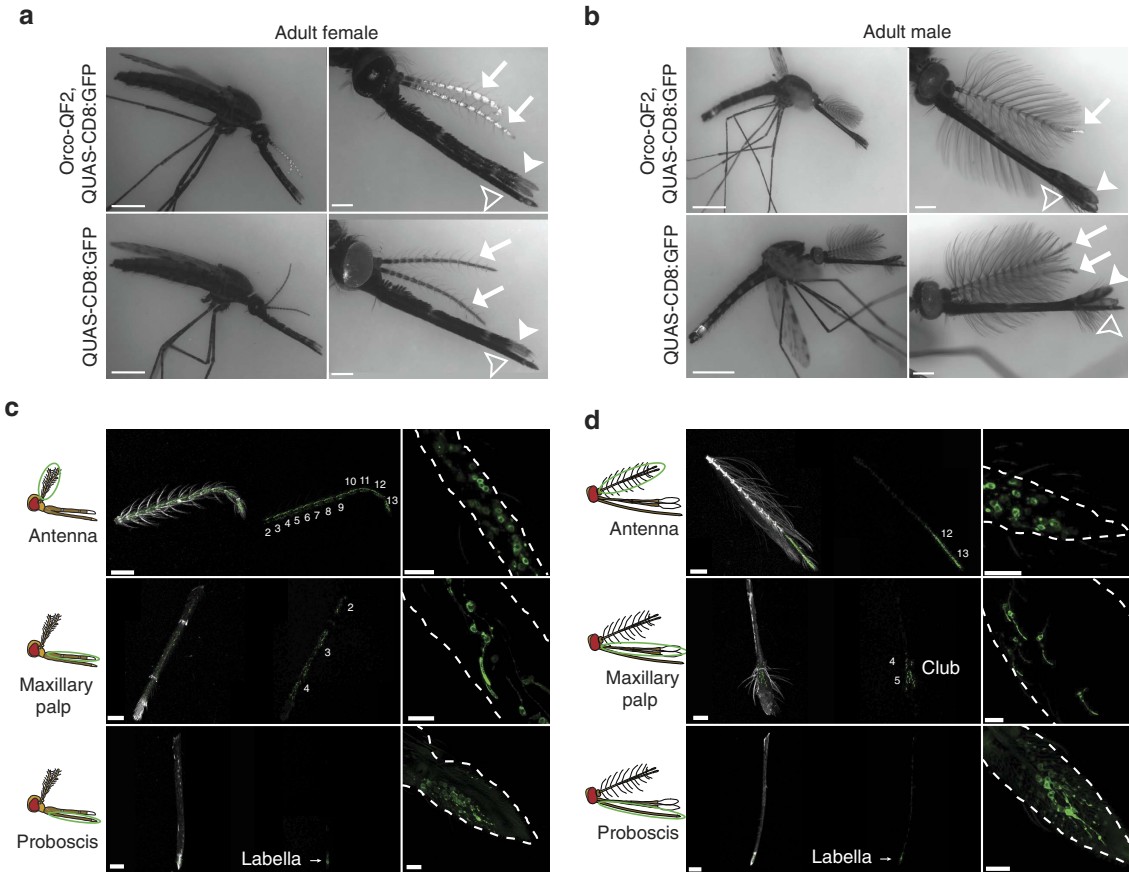

**Figure 1 | *Orco-QF2* expression in olfactory neurons in adult.** (**a**,**b**) Membrane-targeted GFP expression in adult female (**a**, top row) and male (**b**, top row) animals. Bottom row shows images of female (**a**) and male (**b**) *QUAS-mCD8:GFP* controls. GFP expression is visible in antennae (arrows), maxillary palps (white arrowheads) and proboscis (open arrowheads) of transgenic animals (top row), but not in *QUAS-mCD8:GFP* controls (bottom row). Transgenic animals also express fluorescent DsRed and/or CFP markers in the eye, which results in residual fluorescence in the GFP channel. Genotype: *Orco-QF2, QUAS-mCD8:GFP.* Scale bars, (**a**,**b**) left panels, 1 mm; (**a**,**b**) right panels, 200 μm. (**c**,**d**) Confocal images of GFP expression in the antennae (**c**,**d**, top rows), maxillary palps (**c**,**d**, middle rows) and proboscis (**c**,**d**, bottom rows) of female (**c**) and male (**d**) adult animals. Images in **c**,**d** left panels were acquired in RFP (autofluorescence) and GFP channels, which are shown separately for clarity. (**c**,**d**) Right panels are high-magnification images showing the cell bodies of labelled neurons. Scale bars, (**c**,**d**) left panels, 200 μm; (**c**,**d**) right panels, 20 μm.

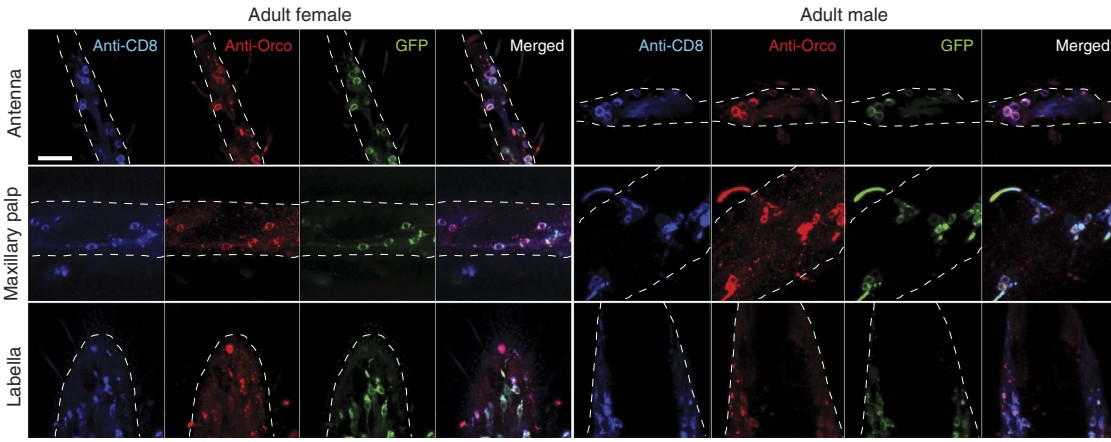

**Figure 2 | *Orco-QF2* drives GFP reporter expression in adult Orco+ neurons.** Antennae (top row), maxillary palps (middle row) and proboscis (bottom row) of adult female (left) and male (right) mosquitoes were immunostained with anti-CD8 (blue) and anti-DmOrco (red) antibodies. Images show immunofluorescence of CD8 and Orco, as well as genetically driven GFP (green). Same cell bodies are labelled in all three channels. White dashed line marks borders of the imaged tissue. All images were acquired at the same magnification. Genotype: *Orco-QF2, QUAS-mCD8:GFP.* Scale bar, 20 μm.

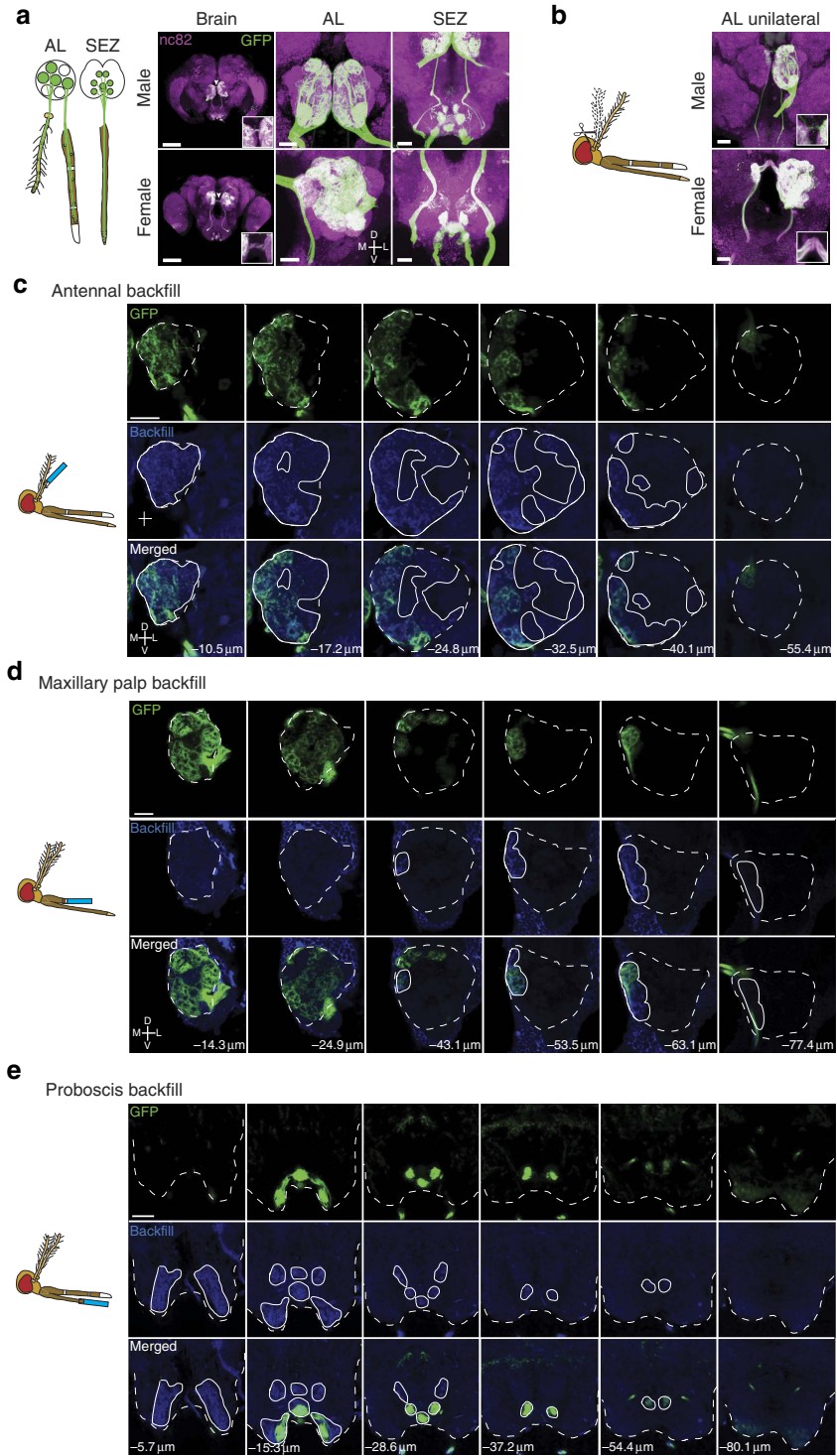

**Figure 3 | Orco+ olfactory neurons target two sensory brain regions.** (**a**) Orco+ receptor neurons send projections to the AL and SEZ of the brain, as detected in *Orco-QF2, QUAS-mCD8:GFP* animals by anti-GFP antibody labelling. Left column shows an overall image of the brain (scale bar, 100 μm). Middle and right columns show the AL and SEZ at higher magnifications (scale bar, 20 μm). Confocal sections are maximum *z*-projections. Arrowhead points to commissure. Inset shows commissure at higher magnification. (**b**) Shown are brains of male and female *Orco-QF2, QUAS-mCD8:GFP* mosquitoes in which the right antennae had been ablated (as schematized in the cartoon) 5 days before brain dissections. The ALs retain ipsilateral Orco+ innervation from the unablated left antennae and bilateral innervation from the intact maxillary palps. Inset shows commissure at higher magnification. Scale bars, 20 μm (**c–e**). Antennae (**c**), maxillary palps (**d**) or proboscis (**e**) of *Orco-QF2, QUAS-mCD8:GFP* mosquitoes were cut to ∼1/3 of their length (as depicted in the cartoons) and backfilled by neurobiotin that is incorporated into the membranes of severed neurons. The neurobiotin labelling (blue) of the brains, together with the GFP labelling (green), establishes the origin of Orco+ receptor neurons (antennal, maxillary palp or proboscis) that innervate the AL and SEZ. Backfills originating from the antenna innervated only the ipsilateral AL. Backfills originating from the maxillary palps innerved both ipsilateral and contralateral ALs. Backfills originating from the proboscis innervated only the SEZ brain region. The numbers indicate the distance in micrometres from the most anterior confocal section. Dashed white lines outline the AL/SEZ. Solid white lines outline the area labelled by the neurobiotin backfill (blue signal). Scale bars, 20 μm.

females than in males (Fig. 3a). To quantify these differences, we performed three-dimensional (3D) modelling of male and female *A. gambiae* brains and antennal lobes (Methods). Female brains ($5.1 \times 10^6 \pm 0.3 \times 10^6 \, \mu m^3$, mean $\pm$ s.e.m., $n = 3$) were 1.07 times larger in volume than male brains ($4.8 \times 10^6 \pm 0.3 \times 10^6 \, \mu m^3$, mean $\pm$ s.e.m., $n = 3$). In contrast, female antennal lobes ($1.7 \times 10^5 \pm 0.1 \times 10^5 \, \mu m^3$, mean $\pm$ s.e.m., $n = 5$) were 1.9 times larger than male antennal lobes ($0.9 \times 10^5 \pm 0.02 \times 10^5 \, \mu m^3$, mean $\pm$ s.e.m., $n = 3$), reflecting increases in individual glomerular volumes. The increase in AL size in females is in agreement with a larger number of Orco + receptor neurons in the antennae and maxillary palps innervating AL glomeruli. Some Orco + neurons project bilaterally, as evidenced by a clearly labelled commissure between the left and right ALs (Fig. 3a,b). Unexpectedly, the SEZ area of the brain is also innervated by Orco + neurons in a pattern of six strongly labelled and two weakly labelled glomerular structures (Fig. 3a). As in *Drosophila* the SEZ region is known to receive gustatory inputs from the proboscis, we hypothesized that Orco + neurons from the mosquito labellum projected to the SEZ.

To determine the origin of the bilaterally projecting Orco + neurons, we ablated the right antennae of adult *Orco-QF2, QUAS-mCD8:GFP* animals and maintained them for 5 days in standard conditions, to allow sufficient time for the axons of the severed olfactory neurons to degenerate, as previously demonstrated in *Drosophila*[48]. The ALs retained Orco + commissural labelling and innervation of only two posterior glomeruli in the right (ipsilateral) AL of the brain, whereas the left AL remained largely unaffected (Fig. 3b). This suggests that antennal olfactory neurons project only to the ipsilateral AL, consistent with previous results[49,50] and in contrast to bilateral projection patterns of *Drosophila* antennal olfactory neurons[48]. The two remaining glomerular targets probably originate from the maxillary palp. To verify and extend these results, we attempted to label severed olfactory neurons from the different olfactory tissues with the anterograde tracing dye neurobiotin[51]. The neurobiotin backfill labelling of severed antennal (Fig. 3c) and maxillary palp (Fig. 3d) neurons indicated that the two bilaterally projecting glomeruli are indeed innervated by maxillary palp neurons. Thus, all antennal Orco + neurons project exclusively to the ipsilateral side of the brain, whereas maxillary palp Orco + neurons innervate the AL bilaterally, consistent with previous backfill studies[49,51]. Neurobiotin backfills of antennae and maxillary palps resulted in staining of all but three posterior glomeruli that were not reliably labelled in every examined brain (see Fig. 4 for details); however, the antennal and maxillary palp backfills did not stain the SEZ. Neurobiotin backfills of the proboscis (Fig. 3e) confirmed that the SEZ is innervated by Orco + neurons from the labella. In contrast to a previous report[52], we observed no neurobiotin + or GFP + projections from the proboscis to the AL.

**Olfactory neuron innervation model of the AL.** Using our neurogenetic approach we identified 67–70 glomeruli in the AL of females and 67–68 glomeruli in the AL of males (Supplementary Table 4). In contrast to *Drosophila*, the appearance of the AL in *A. gambiae* varies markedly among animals (see Methods), which could explain the slight variation in the number of glomeruli we counted. Previous studies[51] reported 60 glomeruli in a female and 61 glomeruli in a male AL, and assigned names to the glomeruli based on their anatomical position. However, previous work assumed that a large part of the anterior AL was occupied by projections from Johnston's organ mechanosensory neurons[51]. This contradicts the transgenic Orco > GFP expression we observe, in which the majority of the anterior glomeruli are

Orco + . Furthermore, we did not observe GFP expression in the Johnston's organ, consistent with a previous study showing a lack of Orco immunoreactivity in this tissue[17]. All together, this suggests that these anterior AL glomeruli do not constitute a mechanosensory centre. To avoid confusion, we have renamed the glomeruli, assigning number 1 to the most dorsal and anterior glomerulus and subsequent numbers to more ventral and posterior glomeruli (Fig. 4a and Supplementary Table 5). We also have made available a 3D reconstruction model of these data for future studies, to allow renaming of these glomerular regions based on specific OR identities (see Methods).

Transgenic GFP expression, combined with the neurobiotin backfill experiments, allowed us to assign each AL glomerulus to one of four classes based on the origin of their respective receptor neurons (antennal or maxillary palp) and on the status of *Orco-QF2, QUAS-mCD8:GFP* labelling (Fig. 4a). We identified 33 Orco + glomeruli in both females and males (Supplementary Table 5), despite the $\sim 2.5$-fold sexually dimorphic difference in the number of Orco + receptor neurons. The majority of the remaining $\sim 50\%$ of glomeruli are likely to be innervated by olfactory neurons that express either an IR or GR receptor.

Five glomeruli—2 Orco + (numbered 45 and 55, blue colour in Fig. 4a) and 3 Orco − (numbered 62, 63 and 64, yellow colour in Fig. 4a)—were reliably labelled by maxillary palp backfills. The two Orco + glomeruli probably correspond to the OR8 and OR28 receptor neurons[22], and at least one of the three Orco − glomeruli are innervated by neurons expressing the $CO_2$-sensing GR22-GR23-GR24 receptor complex[22].

Varying numbers of glomeruli were labelled by antennal backfills (purple and green colours in Fig. 4a), with 3 posterior glomeruli (numbered as 51, 52 and 67, grey colour in Fig. 4a) exhibiting unreliable labelling. This variability in labelling may be explained by weak labelling of posterior glomeruli and also by the fact that neurons originating in the proximal 25–30% of antennae were not labelled in the backfill experiments and may be projecting to these unlabelled glomeruli. This ambiguity will be resolved by future transgenic labelling of neurons that express single receptor types.

The ability to distinguish the olfactory tissue of origin and the Orco-expression status for glomerular targets allowed us to predict a functional map of the AL (Fig. 4b). The organization of glomeruli within the AL is remarkably similar between *Drosophila* and *A. gambiae* (Fig. 4b). Orco + glomeruli are anterior, whereas Orco − glomeruli (IR and GR glomeruli in *Drosophila*) are posterior. This potential evolutionary conservation of AL anatomy suggests that in *A. gambiae*, as in *Drosophila*, most posterior Orco − glomeruli are targeted by IR-expressing neurons. The ventral location of GR + glomerular targets in *Drosophila* is suspected to have coincided with a developmental change during which GR + neurons located in the maxillary palps switched to the antennae[53]. Thus, it is likely to be that the medial–central glomerular location of Orco − , MP + glomeruli in *Anopheles* reflects tissue of origin, as all maxillary palp olfactory neurons in *Drosophila* also target medial–central glomeruli[9,10]. It remains to be explored whether the similarities in olfactory neuronal organization between the two species are also seen in higher brain regions[54].

**Olfactory innervation model of the SEZ.** In the SEZ, we identified eight glomerular structures formed by Orco + neurons originating from the labellum (Fig. 4c). Males and females exhibited similar SEZ innervation patterns. From these confocal images, we generated a 3D reconstruction of the Orco + olfactory neuron innervation pattern in the SEZ and outlined the glomeruli in different colours within the olfactory nerve tracts (Fig. 4d).

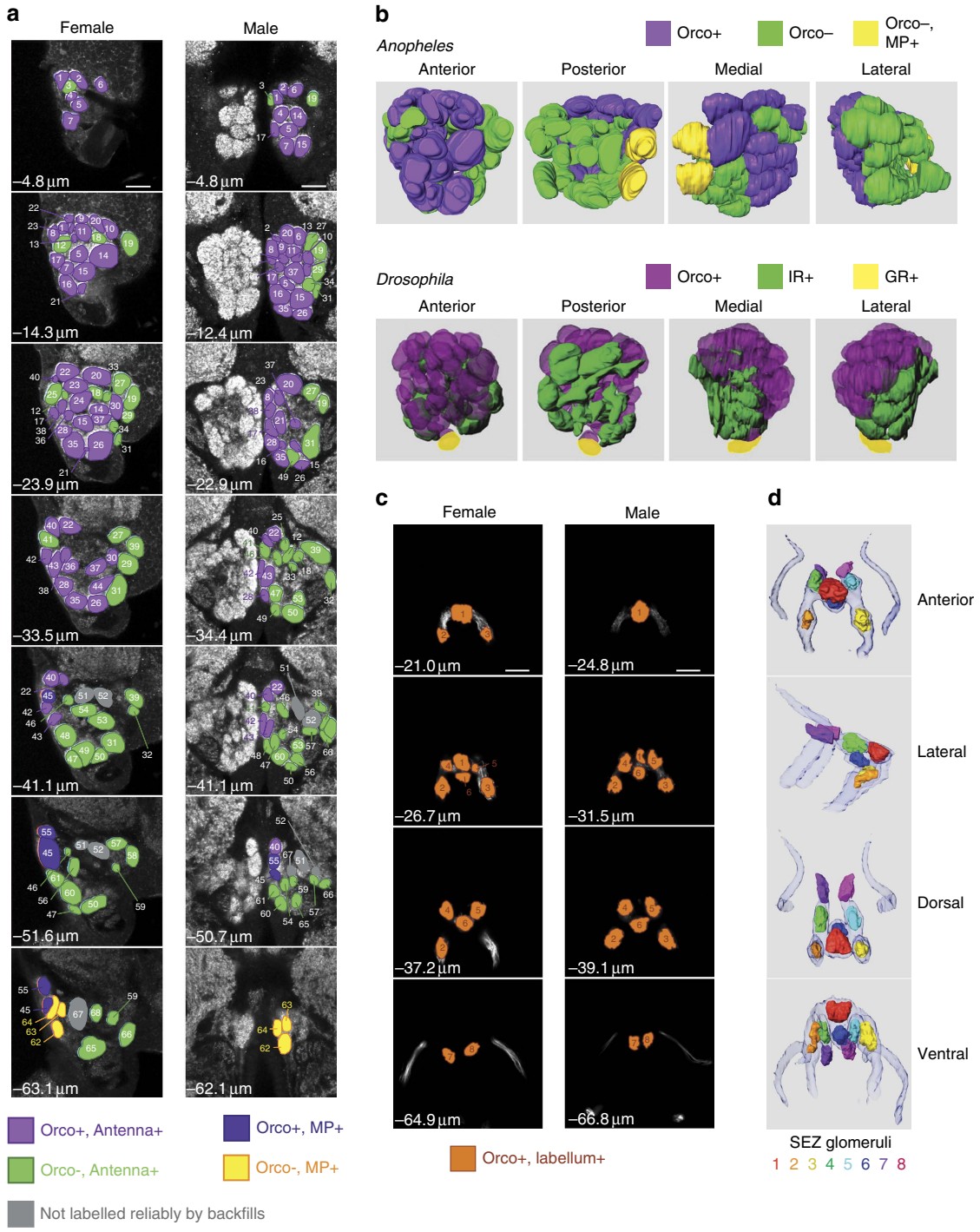

**Figure 4 | Reconstruction of the mosquito AL and SEZ based on Orco expression and olfactory tissue of origin.** (**a**) Confocal Z-stacks of female (left) and male (right) ALs, shown at the same magnification (most anterior—top row, most posterior—bottom row). Glomeruli were outlined manually, and GFP and neurobiotin backfill signals were used to assign each glomerulus to one of five groups: Orco + antennal glomeruli (purple), Orco − antennal glomeruli (green), Orco + maxillary palp glomeruli (blue), Orco − maxillary palp glomeruli (yellow) and glomeruli that were not labelled by backfills (grey). Glomeruli were numbered starting at the most anterior section. Scale bars, 20 μm. Also see Supplementary Table 5. (**b**) A. gambiae and D. melanogaster ALs show similarities in the arrangement of glomeruli targeted by Orco + (purple) and Orco − neurons. Orco − neurons originating from the mosquito antennae are likely to be IR-expressing neurons (IR, green) and Orco − neurons from the maxillary palps are likely to be GR-expressing neurons (GR, yellow)[22]. The Drosophila AL is reprinted with permission from ref. 76 with minor modifications. (**c**) Confocal Z-stacks of female (left) and male (right) SEZ, shown at the same magnification (most anterior, top row; most posterior, bottom row). Glomerular-like structures were outlined manually using the GFP signal. Glomerular structures were numbered starting at the most anterior section. Scale bars, 20 μm. (**d**) Three-dimensional modelling of olfactory neuron targeting in the SEZ. The Orco-QF2-labelled nerve bundle from the labella is shown for reference.

This highlights the relative positions of the eight labeller glomeruli, with three symmetrical pairs of lateral glomeruli and two central glomeruli along the SEZ midline (see Methods for availability of source data). Backfill experiments of the proboscis (Fig. 3) labelled these glomerular structures and also a broader region in the SEZ probably innervated by gustatory and

mechanosensory neurons[52,55,56] (see Methods for availability of source data). As the labellum is in close proximity to host tissue during bloodfeeding[15], this raises the possibility that it functions to integrate sensory signals from host odorants and tastants.

## Discussion

We report here the first neurogenetic labelling of olfactory neurons in mosquitoes. By visualizing the entire neuronal processes of Orco+ olfactory receptor neurons, these studies revealed that two brain regions are targeted by olfactory receptor neurons in this species—the AL and the SEZ. This is the first time, to our knowledge, that insect olfactory receptor neurons have been found to innervate brain regions outside the AL. These Orco+ neurons originate in the labella, a structure that is in close proximity to host skin during bloodfeeding, suggesting that volatiles of the skin or in the blood may activate these olfactory receptor neurons in female mosquitoes. Indeed, recordings from T2 sensilla on the labella indicate activation by low volatility odorants often found on host tissues or in blood[52,57].

The labellum also contains gustatory neurons[52,55,56] that in the *Drosophila* proboscis are known to innervate the SEZ[23]. It is possible that this region in the mosquito SEZ is a sensory integration centre combining olfactory and gustatory signals. In humans, the sensory integration of taste and smell gives rise to a perception of flavor. Whether this region in the mosquito SEZ underlies a 'flavor' centre remains to be explored. Interestingly, Orco+ olfactory neurons were recently identified at the tip of the hawkmoth (*Manduca sexta*) proboscis and found to be involved in evaluating the quality of flowering plants during pollination[58]. Whether the hawkmoth Orco+ proboscis neurons innervate the SEZ and thus implicate a common sensory integration centre between Lepidoptera and Diptera remains to be determined.

The ability to label the entire neuronal process was accomplished genetically by introducing the Q-binary expression system, comprising a QF2 driver line and *QUAS-geneX* reporter line, into *A. gambiae*. Binary expression systems are powerful genetic tools for investigating and manipulating neuronal circuits, and adoption of these reagents into the mosquito system should accelerate studies in neuronal processing of these dangerous insects. In this work, we used the putative *AgamOrco* enhancer+ promoter region to capture the expression pattern of roughly half of all olfactory neurons in *A. gambiae*. Similar approaches could be applied to capture the expression patterns of olfactory neurons not labelled by Orco, such as the IR neurons and the $CO_2$-sensing GR neurons. By using the promoters from the ionotropic co-receptors Ir8 or Ir25, and the promoters from one of the three GR $CO_2$-sensing complex (Gr22, Gr23 and Gr24), it might be possible to generate new QF2 lines labelling these olfactory neuron populations. To this end, we did generate *Gr24-QF2* transgenic lines using a 961 bp putative enhancer+promoter region, but this did not yield expression. In such cases, alternative genetic strategies, pioneered in *Drosophila*, can be used for capturing tissue-specific expression patterns to drive QF2 (ref. 59).

There are two major advantages to using a binary expression system for tissue labelling. The first advantage is amplification of a promoter signal. If a genomic promoter is weak, meaning transcription driven by the promoter results in only few copies of the target gene, it could be difficult to visualize these tissues directly with conventional fluorescent markers. Yet, when a weak promoter expresses a transcriptional activator such as QF2, even low levels of the activator will continue to drive reporter expression. In our system, we used 15 copies of the QF-binding site *QUAS*, which should lead to robust amplification of even weak promoters. This is supported by the fact that

membrane-targeted GFP fluorescence in olfactory neurons using the Orco promoter is visible even in live animals.

The second advantage is versatility. The *Orco-QF2* line, which has been validated here, can be crossed to other *QUAS-geneX* effector lines for a range of studies. In our experience, all *QUAS-mCD8:GFP* insertions appeared to function in *Anopheles*, suggesting that the generation of additional *QUAS-geneX* lines will not be a limiting step. We chose first to generate a membrane-targeted GFP reporter, *QUAS-mCD8:GFP*, for efficient tissue labelling and anatomical studies. This line will also be useful when validating the expression patterns of new QF2 driver lines. Future studies could instead use reporters that monitor neuronal activity, for example, GCaMP6 (ref. 60). Such reagents will allow the visualization of olfactory neuron activity in response to odours in the peripheral organs and also in their glomerular targets in the brain. These types of studies have been used to characterize an 'olfactory code' in *Drosophila*[24,61,62] and can be applied to similar studies in *Anopheles*, to determine the Orco+ olfactory code that distinguishes the odours of humans from those of other animals. Besides monitoring neuronal activity, other *QUAS* effector lines can be used to manipulate or modulate neuronal activity. For example, by using *QUAS-Chrimson*[63,64] or *QUAS-OrX*, olfactory neuron activities could be modulated by red light or odorants, respectively. This approach has yet to be tested in mosquitoes, but it has the potential to provide insights into effective strategies that disrupt olfactory-based host seeking. A rich repertoire of genetic tools developed in *Drosophila* can be adopted for similar uses in mosquitoes[36].

We have generated a map of the adult *A. gambiae* AL. We have identified glomeruli that are targeted by the antennal or maxillary palp neurons and glomeruli that are targeted by Orco+ or Orco− olfactory receptor neurons. This allowed us to predict which glomeruli may be targeted by IR neurons (Orco−, antennal) and which glomeruli may be targeted by the $CO_2$-sensing neurons (Orco−, maxillary palp). A previous study speculated that a large portion of the *A. gambiae* AL comprised mechanosensory targets from Johnston's organ in the pedicel[51]. However, given that these same glomeruli are clearly innervated by Orco+ olfactory neurons, this seems unlikely. Instead, the AL structure is primarily innervated by olfactory neurons from the antennae and maxillary palps. As in *Drosophila*[65], the mosquito antennae may contain sensory sensilla that respond to heat and whose neurons target posterior Orco− AL glomeruli[66].

By speculating which glomeruli are targeted by Orco+ receptor neurons, IR neurons and GR neurons, the *Anopheles* AL map presented here also highlights the glomeruli likely to be activated by different types of odorants. ORs tend to be activated by host body odorants, such as 4-methylphenol, indole or 1-octen-3-ol[11,25], whereas IRs may be activated by acids and amines found in host body odours, such as lactic acid and ammonia[18]. The GR neurons will be activated primarily by carbon dioxide[22,67]. The number of glomeruli innervated by Orco+ receptor neurons and IR neurons is roughly equivalent, suggesting that host seeking behaviours may rely equally on both types of olfactory neuron signalling. This is consistent with studies in *A. aegypti* in which host seeking was not abolished in *Orco* mutant mosquitoes[68].

The number of Orco+ glomeruli in the male and female ALs was the same, despite the sexually dimorphic location and number of Orco+ neurons in the antennae and maxillary palps. A notable sexually dimorphic difference was the reduced size of male glomeruli and ALs: ∼1.9-fold smaller in volume in males compared with females. This probably reflects the reduced number of innervating olfactory neurons in the glomerular circuit. If each female Orco+ glomerulus can be matched to a

male Orco+ glomerulus and each glomerulus is innervated by neurons expressing the same olfactory receptor(s), this suggests the male olfactory system may represent a scaled-down version of the female olfactory system. As such, the male has the potential to respond to all the same odorants that the female responds to, but due to the reduced number of olfactory neurons, with reduced acuity. This is in agreement with transcriptome profiling of the antennae between males and females[45]. When sensillar numbers were normalized between the sexes, the identity and expression levels of ORs between males and females were remarkably similar[45]. The transcriptome study identified 59 AgORs enriched at least twofold in the antennae and maxillary palps (compared with the body)[45]. In contrast, we were able to identify only 33 glomeruli innervated by Orco+ neurons. It is possible that glomeruli in our model might be further subdivided into additional glomeruli based on specific OR innervations. Another possibility is that mosquito olfactory neurons may frequently express more than one OR[69,70]. In particular, *Anopheles* OR genes clustered in the genome (within ~450 bp of each other) are often co-expressed in the same olfactory neuron (for example, *AgOR13*, *AgOR16*, *AgOR17* and *AgOR55*)[69,70]. Including all *Ag* OR gene clusters (VectorBase, *AgamP4*), as many as 17 *Ag* OR genes might be co-expressed in as few as 6 olfactory neuron types. Furthermore, if receptor combinations are sexually dimorphic for each neuron, this could be a mechanism to enable differential response profiles between males and females, while maintaining overall expression of the same set of ORs.

The maxillary palp is a simplified olfactory organ, containing only capitate peg sensilla that house olfactory neurons. Prior electrophysiological studies identified three neurons within each capitate peg sensillum, with two of these neurons expressing Orco and the third expressing the Gr22–Gr23–Gr24 receptor complex and being $CO_2$ sensitive[22]. By ablating one antenna and allowing for the Orco+ olfactory neurons in the antenna to degenerate, we visualized the two glomerular targets for the two Orco+ olfactory neurons originating from the maxillary palp capitate peg sensilla. By using neurobiotin backfills of severed maxillary palp nerves, we identified five posterior glomeruli in the AL that were innervated from the maxillary palp, including the two glomeruli innervated by the Orco+ olfactory neurons. These data suggest that Orco− neurons from the maxillary palp innervate three glomeruli. The capitate peg Gr22–Gr23–Gr24 $CO_2$-sensing neuron must innervate at least one of these three glomeruli. The other two glomeruli will either be innervated by the $CO_2$-sensing olfactory neurons or currently unidentified non-chemosensory neurons. Besides the capitate pegs, the maxillary palp contains two other innervated sensilla—the campaniform sensilla and sensilla chaetica that probably function as mechanoreceptors, proprioreceptors or touch sensors[14,15]. Alternatively, some of these Orco− glomeruli may be innervated by neurons in uncharacterized sensilla that respond to non-mechanical stimuli such as humidity or temperature, as found in the *Drosophila* AL[71].

We also generated the first map of the Orco+ olfactory receptor neuron targets in the *A. gambiae* SEZ. These may represent a previously unreported mode of olfactory signalling independent of the AL in an insect brain. In *Drosophila*, gustatory neuron targeting in the SEZ is diffuse[23]. Interestingly, the Orco+ olfactory neurons in *A. gambiae* appear to form eight glomerular structures in the SEZ region. A number of questions regarding this innervation remain to be explored. Are these olfactory neuron targets also innervated by gustatory neurons? Do all labellar Orco+ olfactory neurons innervate all SEZ glomeruli or do subsets exist based on OR expression? Are these olfactory glomeruli organized to influence gustatory signals such that favourable host odours trigger pleasant tastes and unfavourable host odours trigger bitter-like tastes?

As vectors for viral and parasitic diseases, anthropophilic insects pose a major threat to human health. Mosquitoes are particularly dangerous due to their ability to visit multiple people during a bloodmeal, such that infections can rapidly spread through both human and mosquito populations. Effectively controlling the spread of mosquito-borne diseases will probably require a multitude of approaches. Given that each step in host seeking—from long-range guidance to short-range biting—involves cues from the olfactory system, targeting this sense could provide an effective means of reducing biting and the spread of disease. The neurogenetic tools introduced here may help identify which aspect of olfactory host-seeking, such as labellar probing or long-range guidance, is the most vulnerable to interference, and thereby guide new strategies for olfactory-based interventions.

## Methods

**Recombinant DNA construction.** Plasmids were constructed by enzyme digestions, PCR, subcloning and the In-Fusion HD Cloning System (Clontech, catalogue number 639645). Plasmid inserts were verified by DNA sequencing.

The pXL-BACII-ECFP-15xQUAS_TATA-mCD8-GFP-SV40 construct was used to generate transgenic animals, carrying a reporter transgene *QUAS-mCD8-GFP*. The *15xQUAS* sequence was PCR amplified from *pQUASp* plasmid (Addgene #46162) using oligos *pBac-ECFP-15xQUAS-Inf-FOR* (5′-aga gcg gcc gcc acc gcg gtc acg tgt cac tgg gtc ag-3′) and *pBac-ECFP-15xQUAS-Inf-REV* (5′-tac cgt cga cct cga gct agc agg tcc tca ctg agt ccc aac gtg aaa g-3′) and InFusion-cloned into *pXL-BacII-ECFP* plasmid[72], digested with XhoI and SacII. Next, the *TATA-mCD8-GFP-SV40-SV40* sequence was PCR amplified from *pQUASt-mCD8-GFP* plasmid[34]; Addgene #24351) using oligos pBac-ECFP-GFP-Inf-FOR (5′-act cag tga gga gga cct gaa ttc ctg cag ccc g-3′) and *pBac-ECFP-GFP-Inf-REV* (5′-tac cgt cga cct cga gag atc tag gcc ttc tag tgg atc cg-3′), and InFusion-cloned into the inter-mediate step construct, obtained at the previous step, that was digested with PpuMI and XhoI.

The pXL-BacII-15xQUAS_TATA-SV40 construct may be used to easily generate new *QUAS* reporter plasmids by sub-cloning a reporter gene into the multi-cloning site between *TATA* promoter and *SV40 terminator* sequences. To create *pXL-BacII-15xQUAS_TATA-SV40*, *pXL-BACII-ECFP-15xQUAS_TATA-mCD8-GFP-SV40* was digested with XhoI and EcoRV-HF to remove the *CD8-GFP-SV40-SV40* cassette. The *SV40 terminator* was PCR amplified from *pQUASt-mCD8-GFP* (ref. 34; Addgene #24351) with oligos *pBac-15xQUAS-SV40-Inf-FOR* (5′-gcg gcc gcg gct cga gac gtc gat ctt tgt gaa gga acc tta ctt ctg-3′) and *pBac-15xQUAS-SV40-Inf-REV* (5′-ttt ctt gtt ata gat atc gat cca gac atg ata aga tac att gat gag-3′) and InFusion-subcloned back into the construct.

The pXL-BACII-DsRed-QF2-hsp70 construct may be used to easily generate new QF2 (ref. 44) driver plasmids by sub-cloning an enhancer/promoter sequence upstream of *QF2*. To create *pXL-BACII-DsRed-QF2-hsp70*, the *QF2-hsp70* cassette was PCR amplified from *pattB-syn-QF2-hsp70* (ref. 35; Addgene #46115) with oligos *pBAC-DSred-QF2-Inf-FOR* (5′-atc aat gta tct cga ggc cgg cca aca tgc cac cca ag-3′) and *pBAC-DSred-QF2-Inf-REV* (5′-ttt ctt gtt ata gat atc gga tct aaa cga gtt ttt aag c-3′) and InFusion-cloned into *pXL-BACII-DSred*[73], digested with XhoI and EcoRV-HF.

The pXL-BACII-DsRed-OR7_9kbProm-QF2-hsp70 construct was used to generate transgenic animals, carrying an *Orco-QF2* driver construct. A 9,312 bp gene immediately upstream of *AGAP002560* (*OR7*) was PCR amplified with oligos *Agamb-OR7-Inf-FOR9_1* (5′-atc aat gta tct cga gtg atg caa att gtt cgg aag aat tg-3′) and *AgambOR7-Inf-Rev1* (5′-tgg gtg gca tgt tgg ccg gcc tgc gaa cgg gaa gtg aac-3′) from genomic DNA, extracted with the DNAeasy Blood & Tissue Kit (Qiagen, #69506) from the Keele strain of *A. gambiae*. The PCR product was InFusion cloned into the *pXL-BACII-DsRed-QF2-hsp70* vector, digested with XhoI and FseI.

**A. gambiae transgenics.** The *A.gambiae* M-form strain Ngousso (the M-form of *A. gambiae* is now referred to as *Anopheles coluzzii*) were grown at 28 °C, 70–75% relative humidity, 12 h light/dark cycle. Freshly deposited eggs were collected by providing mated, gravid females with wet filter paper as an oviposition substrate for 15–20 min, after which the eggs were collected and systematically arranged side-by-side on a double-sided tape fixed to a coverslip. Aligned embryos were covered with halocarbon oil (Sigma, series 27) and injected at their posterior pole with an injection cocktail between 30–40 min after egg laying. Injection cocktails consisted of a mixture of two plasmids, one with a piggyBac vector carrying the transgene of interest with a dominant visible marker gene, DsRed or enhanced cyan fluorescent protein (ECFP), under the regulatory control of the 3xP3 promoter, and a piggyBac transposase-expressing plasmid consisting of the transposase open reading frame under the regulatory control of the promoter from the *A. stephensi vasa* gene.

Vector concentrations were at either 35, 75 or 150 ng μl$^{-1}$, whereas the transposase-expressing plasmid was at 300 ng μl$^{-1}$ in 5 mM KCl, 0.1 mM sodium phosphate pH 6.8. Halocarbon oil was immediately removed and coverslips with injected embryos were placed in trays of water at 28 °C, where the first instar larvae hatched ∼ 24 h later. The Insect Transformation Facility (https://www.ibbr.umd.edu/facilities/itf) within the University of Maryland College Park's Institute for Bioscience and Biotechnology Institute performed all embryo microinjections. Adults developing from injected embryos were separated by sex before mating and small groups of five to ten injected adult males and females were mixed with wild-type Ngousso adults of the opposite sex. The progeny from these matings were screened during the third or fourth larval instar for the presence of vector-specific marker gene expression. Transgenic larvae were saved and as adults were backcrossed to Ngousso (wild type).

Using the strategy of promoter cloning as described for *Orco*, we also generated *promoter-QF2* transgenic *Anopheles* using predicted promoter sequences for *GR24* (AGAP001915; 957 bp), *elav* (AGAP001883; 6,569 bp) and *n-syb* (AGAP005507; 10,194 bp). These *QF2* transgenic lines did not show expression when crossed to *QUAS-mCD8:GFP*, suggesting the promoter fragments used were insufficient for capturing the expected expression pattern.

**A. gambiae stock maintenance.** *A. gambiae* were grown at 28 °C, 70–75% relative humidity and 12 h light/dark cycle. Larvae were provided with TetraMin Tropical Flakes and/or Purina Cat Chow Indoor pellets ad libitum, whereas adults were provided with 10% sucrose continuously. Mated, adult females were fed human blood using temperature-controlled membrane-feeding devices or mouse blood from anaesthetized mice and eggs were collected from the resulting gravid females by providing them with a cup of water containing wet filter paper on which to deposit their eggs. *Orco-QF2* and *QUAS-CD8:GFP* transgenes were brought together by a genetic cross and maintained in the same stock for at least seven generations with no observable reduction in the strength or breadth of GFP expression.

**Immunohistochemistry.** Dissection of adult brains and immunostaining were done as described previously[74]. In short, on day 1 heads of adult mosquitoes were fixed in 4% paraformaldehyde in 0.1 M sodium phosphate buffer (Millonig's pH 7.4) with 0.25% Triton X-100 at 4 °C for 3 h. Next, the brains were dissected in PBS containing 0.3% Triton X-100 (PBT), washed in PBT for 1 h at room temperature (RT) and permeabilized with 4% Triton X-100 + 2% normal goat serum (NGS) in PBS at 4 °C overnight. On day 2, the brains were washed in PBT for at least 1 h and placed in primary antibody mix + 2% NGS in PBT for two nights at 4 °C. On day 4, the brains were washed for several hours in PBT at RT and placed in secondary antibody mix + 2% NGS in PBT for two nights at 4 °C. On day 6, the brains were washed in PBT and placed in mounting solution (Slow Fade Gold) overnight at 4 °C, and mounted on a microscope slide on day 7. To visualize CD8:GFP expression, we used rat anti-CD8 (Invitrogen #MCD0800, 1:100) and mouse nc82 (DSHB, 1:50) primary antibodies, and Alexa-488 goat anti-rabbit (Invitrogen #A11034, 1:200) and Cy3 goat anti-mouse (Jackson ImmunoResearch #115-165-062, 1:200) secondary antibodies.

Anterograde tracing experiments with neurobiotin were performed essentially as described in ref. 51. Live mosquitoes were cooled in a 4 °C fridge for 10–15 min, after which they were transferred to a petri dish and placed on ice for up to 30 min until they were mounted in the backfill chamber. The backfill chamber consisted of a 9 cm petri dish with two parallel strips of Blue-Tack (Bostik, France) on the bottom of the dish, 2 cm apart. Double-sided sticky tape was placed on top of one of the Blue-Tack strips. The other strip was used to hold glass pipettes, which were pulled out of glass capillaries (Harvard Apparatus, #30–0108), and had the tips broken off to create an opening of ∼ 0.3 mm. The pipettes were backfilled with 2% Neurobiotin (Vector Laboratories, # SP-1155 or #SP-1150) in 0.25 M KCl. Mosquitoes were mounted on the double-sided tape and either one antenna, one maxillary palp or the proboscis was cut to ∼ 25% of the original length and inserted into the tip of the biotin-filled glass pipette. We used a separate pipette for each animal. Small pieces of damp Kimwipes were placed into the backfill chamber, to prevent desiccation of the animals. The chambers were wrapped in aluminum foil and placed at 4 °C for 20–24 h, after which we followed the Brain immunostaining protocol, described above. When Neurobiotin-Plus (Vector Laboratories, # SP-1150) was used for backfills, Avidin-Texas Red Conjugate (Molecular Probes, #A-820, 1:6,000) was added to the primary antibody mix and 647 goat anti-mouse (Life Technologies #Z25008, 1:200) secondary antibody was used instead of Cy3 goat anti-mouse. When Neurobiotin-350 (Vector Laboratories, # SP-1155) was used for backfills, no modifications were required for the Brain immunostaining protocol. Brains were kept in the dark at all times.

Whole-mount preparation of adult antennae, maxillary palps and proboscis was performed as described in ref. 70 with minor modifications. On day 1, mosquitoes were immobilized on ice, their heads were cut off and placed in ZnFA fixative solution (0.25% ZnCl$_2$, 1% formaldehyde, 135 mM NaCl, 1.2% sucrose and 0.03% Triton X-100) in 250 μl PCR tubes for 20–24 h at RT in the dark. On day 2, the heads were washed twice for 15 min each with HBS buffer (150 mM NaCl, 5 mM KCl, 25 mM sucrose, 10 mM Hepes, 5 mM CaCl$_2$ and 0.03% Triton X-100). Appendages were carefully cut off in HBS and placed in separate PCR tubes, based on gender and type of tissue (six PCR tubes in total for antennae, maxillary palps

and proboscises of male and female mosquitoes). After a brief wash in HBS, the tissue was incubated in 80% methanol/20% dimethyl sulfoxide (DMSO) solution for 1 h at RT, washed for 5 min in 0.1 M Tris pH 7.4, 0.03% Triton X-100 solution and incubated in blocking solution (PBS, 5% NGS, 1% DMSO and 0.3% Triton X-100) for at least 3 h at RT. Next, the tissue was put in blocking solution containing the primary antibodies, the PCR tubes were placed in a buoyant perforated plastic pipette tip holder and allowed to float in a water bath sonicator (Bansonic 1,200, Branson, Danbury, CT) for 5 min (antennae and maxillary palps), or held with forceps and dipped into the water bath sonicator for 30–60 s (proboscises). Next, the PCR tubes were placed for 4 days at 4 °C in the dark. Sonification was repeated on day 3. On day 6, the tissue was washed at RT in PBS, 1% DMSO and 0.3% Triton X-100 for 2–3 h. Secondary antibodies were added to blocking solution, the tubes were sonicated as described above and incubated for 3 days at 4 °C in the dark. On day 9, the tissue was washed at RT in PBS, 1% DMSO and 0.3% Triton X-100 for 2–3 h, rinsed in PBS and mounted in Slow Fade Gold. To visualize CD8:GFP expression, we used a rat anti-CD8 primary antibody (Invitrogen #MCD0800, 1:100). To visualize Orco protein, we used a rabbit anti-DmOrco antibody[29] (gift from Leslie Vosshall, Rockefeller University, 1:100). The secondary antibodies were Cy3 goat anti-rabbit (Jackson Immuno Research #111-165-144, 1:200) and 633 goat anti-rat (Invitrogen #A21094, 1:200). We did not use any green secondary antibody, to clearly visualize endogenous GFP fluorescence, driven by the transgenic reporter.

Whole-mount preparation of larval antennae and maxillary palps was performed similar to the adults. On day 1, third and fourth instar larvae were separated by sex (females with red markings on their cuticle, males with pale cuticles; larvae with ambiguous markings were excluded). The larval heads were cut off and placed in ZnFA fixative solution in two separate 250 μl PCR tubes (one for males and one for females) for 20–24 h at RT in the dark. The following steps were performed identically to the adult antennae and maxillary palps, with one exception: larval appendages remained attached to the heads throughout the staining protocol and were only dissected off immediately before mounting.

Cryosections of adult antennae were performed as described in ref. 17. On day 1, heads with attached antennae of female mosquitoes were dissected and pre-fixed in 4% paraformaldehyde in.03% PBT at 4 °C for 30 min, then rinsed three times with PBT and placed in a 25% sucrose solution in PBT overnight at 4 °C. On day 2, samples were submerged in Tissue-Tek OCT compound (Sakura Finetek, Torrance, CA) and frozen at − 80 °C before being sectioned at 12 μm on a Microm HM 500 M cryostat (Microm International GmbH, Walldorf, Germany). Sections were collected on SuperFrost Plus slides (Fisher Scientific) and dried for 30 min at RT, then fixed in 4% paraformaldehyde in.03% PBT at RT for 30 min. Slides were rinsed three times with PBT at RT, blocked in PBT + 5% NGS for 1 h at RT and incubated with primary antibody in blocking solution overnight at 4 °C (rabbit anti-DmOrco[29], gift from Leslie Vosshall, Rockefeller University, 1:100). On day 3, slides were rinsed three times at RT with PBT, then incubated with secondary antibody in block for 2 h at RT (Cy3 goat anti-rabbit, Jackson Immuno Research #111-165-144, 1:200). No green secondary antibody was used, to visualize endogenous GFP fluorescence. Slides were rinsed three times with PBT at RT, then mounted in Slow Fade Gold.

**Whole-animal imaging.** Adult mosquitoes were anaesthetized at 4 °C, placed on the ECHO therm chilling/heating plate (Torrey Pines Scientific) set to 0 °C and imaged by a Zeiss SteREO Discovery.V8 microscope equipped with a GFP-470 and ds-Red filters and a Jenoptik ProgRes MF cool charge-coupled device camera. Images were acquired in ProgRes Mac Capture Pro 2.7 software and stored in *.tif format. Images that are compared with each other were obtained under identical hardware and software settings.

Petri dishes filled with water containing larval mosquitoes were placed on the ECHO therm chilling/heating plate set to 0 °C and imaged as described for the adults.

**Confocal imaging and analyses.** Brains, head appendages and cryosections were imaged on a LSM 700 Zeiss confocal microscope at 512 × 512 pixel resolution, with 0.96 μm or 2.37 μm Z-steps. For illustration purposes, confocal images were processed in ImageJ to collapse Z-stacks into a single image; in some cases, the colour channels were swapped or duplicated.
No other image processing was performed on the confocal data.

To quantify Orco and mCD8-GFP co-expression in the adult (Supplementary Table 3), we used the 3D reconstruction software Amira (FEI, Oregon, USA) to manually outline cells stained with anti-Orco or anti-CD8 antibodies. Each outlined cell was assigned a separate label. Cells were marked independently in Orco (red) and in CD8 (blue) channels, creating two layers of labels in Amira. Next, 3D surfaces, corresponding to the outlined cells, were generated by the software and manually inspected as to the overlap of Orco and CD8 channels.

To quantify Orco (red) and mCD8-GFP (blue) co-staining with GFP fluorescence (green) in the larvae (Supplementary Tables 1 and 2), we used ImageJ with the Image5D plugin and Zen Zeiss software to mark and count cells in each of the three fluorescence channels.

To quantify the overlap of Orco staining (red) and endogenous GFP fluorescence (green) in the cryosectioned female antennae, we used ImageJ with the

Image5D plugin and Zen Zeiss software to mark and count cells and sensilla in each channel.

The 3D reconstructions of the AL and SEZ were generated in Amira as described above. For the AL, glomeruli were manually identified and labelled in the nc82 channel, and coloured based on whether they were also labelled in the Orco channel, the backfill channel or neither of these two. The correspondence between the data, presented in this study and in ref. 51 (Supplementary Table 5), was established by visual inspection of images in ref. 51 and finding a glomerulus in our data that appeared to have the closest anatomical location to each reported glomerulus. For the SEZ, glomerular-like structures were identified and labelled in the GFP channel, and volume rendering of the same channel was used to show processes. The 3D models are available (see below).

ALs showed high variability in appearance between individuals, in contrast to what we have observed in Drosophila[75]. Experimental conditions (for example, age, feeding and growth) were maintained as much as possible. Nonetheless, differences between AL samples probably arose due to a combination of technical (dissections and mounting) and biological reasons (for example, a rigid oesophagus pressing against ALs deforming glomeruli to different degrees). Future transgenic labelling of individual olfactory neuron classes will help address the source of glomerular variability.

Adult brain volumes were calculated as follows: we manually traced each confocal plane of nc82-stained male and female central brains in Amira (excluding optic lobes) and used the Amira software to generate 3D surfaces from the traces. These were used to calculate the enclosed surface volume of the central brain. AL volumes were calculated by summing 3D reconstructions of each glomerulus for each AL.

**Data availability.** The authors declare that data supporting the findings of this study are available with the article (and its Supplementary Information files). Source data and 3D models shown in Fig. 4a,b,d are available from the Potter Lab website at Johns Hopkins University School of Medicine (http://potterlab.johnshopkins.edu/natcomm2016). Additional data or information pertaining to this work are available from the corresponding author upon request.

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

## Acknowledgements

This work was supported by a Johns Hopkins Medicine Discovery Fund and a Johns Hopkins Malaria Research Institute Pilot Grant. R.A. and D.O. were supported by a grant from the NIH NIAID (R01AI099060). We thank Leslie Vosshall for the anti-DmOrco antibody, Marcelo Jacobs-Lorena, Alex Kolodkin, Meg Younger and Conor McMeniman for discussions, and the Insect Transformation Facility (Rockville, MD) for *Anopheles* embryo injections. We thank Winter Okoth and Christopher Kizito at the Johns Hopkins School of Public Health for technical help with rearing mosquito lines.

## Author contributions

O.R., D.T., C.-C.L. and E.M. performed the experiments and analysed data. R.A. and D.A.O. produced transgenic mosquitoes. O.R. and C.J.P. wrote the manuscript with input from all authors. C.J.P. supervised the project.

## Additional information

**Competing financial interests:** The authors declare no competing financial interests.

