## [Peer Review File · Nature Communications]

Reviewer #1 (Remarks to the Author)

This is an interesting study that describes for the first time the use of the Q-system of binary expression in the mosquito. It is used here to visualize ORNs in *Anopheles*. The study produces a map of antennal lobe glomeruli that should be very useful to the field. Overall this is an impressive study that makes an important advance in the field of vector biology.

1. The paper should cite the work of Sylvia Anton on the projections of ORNs in the antennal lobe of *Anopheles*.
2. p. 7 Was there weak, un-induced expression of the reporter in the MB and optic lobe neurons in both of the QUAS-mCD8:GFP transgenic lines?
3. p. 8 There are twice as many ORNs as Orco+ cells. The authors suggest that many sensilla contain no Orco+ cells. Another possibility is that many sensilla contain one Orco+ and one Orco-cell. Do the authors know from the labeling patterns whether most labeled sensilla contain one or two labeled cells?
4. p. 9 GFP fluorescence was observed in the hindgut. Was there any in the male reproductive track? Zwiebel's group has reported expression of Ors in *Anopheles* sperm (PNAS, 2014).
5. p. 10 "clearly labeled commissure between the left and right ALs." It would be helpful to indicate these with an arrow in both males and females. Also, the male and female images in Fig. 3a appear very different and this difference needs to be explained to those who are not familiar with antennal lobes.
6. p. 10 Antennal ORNs project only to the ipsilateral side. Maxillary projections are bilateral.
--The work of Anton and others should be cited here.
--The *Anopheles* antenna seems to differ from the *Drosophila* antenna. The authors should discuss additional species so readers will know whether one of these two species is an outlier.
7. p. 11 in contrast to *Drosophila*, the appearance of the AL varies markedly among animals. This deserves comment and discussion; the reader will wonder whether this interspecific difference arises from biology (e.g. more genetic variability in the mosquito population) or technical considerations.
8. p. 12 ... "this implies that the same olfactory receptors may be expressed in the animals of both genders...." The logic here is not clear. The results discussed in this paragraph would seem to be compatible with a model in which different Ors are expressed in males and females. (There is no citation in this para. for genomic analysis.) Also, it's not clear to the casual reader why the assumption of one Or/ORN needs to be invoked here.
9. p. 14 "... the labellum....might also be used to identify short-range host volatiles released during a successful bite. If such volatiles are not detected, the mosquito continues probing." How often does *Anopheles* bite the same individual twice in rapid succession? A reference would be helpful here.
10. Fig. 4A key: it's difficult to distinguish the darker blue from the lighter blue that is supposed to designate the "not labeled" glomeruli.
11. Figs. 3cd needs better explanation - especially because the backfill labeling is very dim. Perhaps arrowheads would help.

Reviewer #2 (Remarks to the Author)

The work of Riabinina and colleagues is a straightforward development and analysis of transgenic tools in the mosquito *Anopheles gambiae* to visualize the subset of olfactory receptor neurons (ORNs) that express the OR coreceptor ORCO. The technical innovation in applying the QF bipartite expression system to *A. gambiae* is very nice (although not completely ground-breaking; for example, the Gal4/UAS system has already been established in this species, as acknowledged by the authors). Nevertheless, the exploitation of their tools to document the anatomical properties of ORNs helps confirm, correct and extend previous histological studies in this species, which relied entirely on the (imperfect) neuronal backfill method to relate peripheral chemosensory organs to central projection domains. Although the expected readership is probably relatively limited (mosquito and other insect olfactory researchers), it is nevertheless an important piece of work for bringing cutting-edge genetic tools into ecologically and biomedically-relevant species.

My main concern is the surprising observation that only 33 glomeruli are labeled with ORCO-QF, because this is inconsistent with previous expression data (from the Zwiebel lab) indicating that about twice as many ORs are expressed in the antenna. As the authors mention, this either implies a departure from the 1 OR:1 ORN rule, or that their defined glomeruli boundaries actually represent more than one glomeruli, but I find both of the explanations unlikely (at least as the necessary high frequency to make the numbers match). Also, the innervation of many trichoid sensilla with non-ORCO positive ORNs would be different from all other characterized insect species, which is possible but surprising. In addition, this unexpectedly low number of ORCO glomeruli also leaves roughly 40 "vacant" glomeruli, which the authors suggest would be innervated by IRNs. However, the Zwiebel lab data suggest that only about half this number of IRs are expressed in the antenna (i.e., not enough for occupying all these glomeruli). These discrepancies make me wonder whether the ORCO-QF line is as faithful as the authors argue based upon quantification of ORCO/CD8 co-staining. If it is true that antibody staining on whole mount is not entirely reliable, is it possible that they authors may have missed visualizing a large number of endogenous ORCO-positive cells that do not express the ORCO-QF. If I understand their quantifications, they see staining of, for example, 35-70 ORCO/CD8 positive cells per female antenna, but this only a very small fraction of the 500-600 GFP labeled cells (visualized without staining). (This contrasts with larvae, where it appears that there is an excellent correspondence between anti-ORCO, anti-CD8, and GFP+ cells). If the adult appendages are understained, then more thorough validation experiments are needed (for example, by staining antennal sections, where antibody penetration is less likely to be problematic) because the faithfulness of the ORCO-QF line is key to many of the observations and interpretations.

Minor comments:

- it would be interesting to confirm or refute the hindgut expression of the ORCO-QF line with antibody staining (rather than leaving this only as a provocative, but potentially artifactual, expression pattern).
- the authors makes observations counter to previous work from Zwiebel (regarding labellar projections to the antennal lobe) and Ignell (whose proposed mechanosensory center is actually composed of ORCO-positive neurons). Could they elaborate on the basis for the previous observations and why they observe different results?
- the Introduction and Discussion could be shortened. There is quite laborious discussion of the background to quite standard bipartite gene control systems, and, in particular, the health impact of mosquito vector research, which seems unnecessary and/or irrelevant to the present manuscript. As one example, lines 323-337 could be removed without losing any impact on this manuscript; it's fine to mention the idea of a flavor center but without any functional data it remains speculative.
- figure 1a - are the scale bars correct? (notably the "100 μm" scale bars)

- in the co-expression analyses, it would be helpful to show the merged as well as the separate channels.

Reviewer #3 (Remarks to the Author)

Riabinina et al. present a major technical breakthrough in neurogenetics by establishing the bipartite QF/QUAS transcriptional activation system in the malaria mosquito, *Anopheles gambiae*. This is the first time that the Q system has been shown to work outside of the conventional model organisms, and this paper opens up an entire field of neural circuit analysis in this important vector mosquito. Here the authors label all cells expressing the olfactory receptor co-receptor (*orco*) with a membrane-tethered GFP and examine the neuroanatomy of *orco* in larvae and adults. While most of what was discovered was expected based on earlier anatomical work in the mosquito, and older work in *Drosophila melanogaster*, there are a few surprises. Although female mosquitoes have more than twice as many *orco*-positive olfactory neurons than males, the number of antennal lobe glomeruli is comparable between the sexes. *Orco*-positive fibers innervate the subesophageal zone, which has not previously been shown to receive olfactory input. Overall the work is technically sound, and opens up a new field of neurogenetics in the mosquito. The work would benefit from some revisions of text and figures as detailed below.

MAJOR POINTS

1. The title and abstract emphasize the organization of olfactory centers, but the work is as much or more about the Q system working in the mosquito. To my taste the anatomical findings about the organization of sensory centers in the adult brain are solid, but down the road this paper will be read and cited because it is the first use of QF/QUAS in mosquitoes. That should be made clearer in the title and abstract.

2. At three places in the paper, the authors provide extensive speculation that I feel goes too far beyond the data to support it.

The first is the assertion that most projections are ipsilateral, but a subset are contralateral. The ablation experiments are suggestive, but not definitive, as the kind of clonal analysis done routinely in *Drosophila* would be. The images provided are not helpful for the reader to judge the evidence for contralateral projections. I suggest backing off a bit from the claims, and put in images that show the commissure.

The second point is the assumption that the "missing" glomeruli in the antennal lobe must express gustatory receptors (GRs) or ionotropic receptors (IRs). The authors present this as a foregone conclusion, and while it is certainly likely, the mosquito is already showing us some surprising results that differ dramatically from the situation in the fly. So I would not go so far as to call the circuit solved based on assumptions that the mosquito and fly antennal lobe will be homologous.

The third point is the evolutionary arguments that compare the mosquito and fly antennal lobe, specifically the numerology of the presumptive carbon dioxide sensitive glomeruli. The authors interpret their staining to be 3 separate glomeruli, and suggest that this might receive different types of activity. There is not enough evidence in the pictures alone to conclude this. In fact, there is not evidence presented that these are indeed the carbon dioxide glomeruli. Again, a more nuanced discussion that allows some uncertainty is recommended. It would be nice to go beyond just comparing the anatomy between mosquito and fly, perhaps to the moth, honeybee, and ant glomeruli for which there are 3-D atlases.

3. A few interesting puzzles emerge from the results, that I would like to see the authors comment on. The first is the observation that female mosquitoes have dramatically more sensory neurons but a similar number of antennal lobe glomeruli. Are the male glomeruli much smaller than the female? If not, do the authors have any thoughts about how the male brain is compensating for much reduced input? The second is why are there only 33 *orco* glomeruli and 79 ORs? A discussion of this mismatch would be interesting.

4. The authors introduce the concept that the mutual innervation of *orco* neurons of the antennal lobe and the SEZ is evidence for smell/taste fusion, or the emergence of "flavor." This is an

important and interesting concept, but my sense is that not all the data are in yet to draw this strong psychophysical conclusion.

5. Because larval expression is not integral to the paper, Figure 1a and all discussion of larval expression could be removed from the manuscript. A future analysis of the larval chemosensory system is a better place to feature such data.

FIGURE SUGGESTIONS

Figure 1 Given the bleed-through of 3xp3 and the scattered constitutive expression of QUAS-CD8-GFP heterozygote with no driver, QUAS-CD8-EGFP and not wild-type should be the control in these staining experiments.

Figure 1: Beyond quantifying the number of ORCO+ cells in each antenna, the authors could take this one step further and quantify the number of ORCO+ cells per segment. This might reveal something about the uniformity of ORCO expression, and suggest the composition of each segment as an identical repeating unit or not.

Figure 2, and Figure 3c-e: in addition to the red, green, and blue panels, please provide merged views of all samples so that it is possible to compare the staining in a single image.

Figure 4a, b: use the same color scheme for marking the glomeruli according to orco gene expression in both panels a and b. Currently it is quite confusing to go from one panel to the next.

Figure 4: A major difference in this work relative to previously published work is that the JOC is not mechanosensory but olfactory. If the authors have any insights in where it projects, perhaps by dye-filling the JOC, that would be helpful. Does it go to AMMC or does it also send projections to these ORCO+ glomeruli?

Figure 4c: the green shading is not visible on the actual images, please check or select another shading color.

Minor points:

1. L36 remove "SOG" as an abbreviation, it has been superseded by "SEZ" per the Kei Ito neural nomenclature system.
2. L189 word choice, gender should probably be "sex"
3. L214 I suggest that the statement about "high background" of the anti-orco antibody be put into context. The antibody (generated by my lab) is selective for orco in *Drosophila melanogaster* and *Aedes aegypti*, as verified by the absence of staining in the null mutants generated in each species. Without an orco null mutant in *Anopheles gambiae*, it is hard to state for certain that the staining is background or signal. A more cautious disclaimer is needed here.
4. L257 typo positive
5. L350-353 other groups will be interested in whether any other transgenes were attempted that did not give expression.
6. The Figure legend for Figure 1b refers to larvae, whereas I think the images are from adult antenna.
7. State in the manuscript that the antennal lobe atlas will be available for others to download and work with.

Please note that I no longer participate in anonymous peer review, and ask that my name be associated with this review and be made known both to the authors and the other reviewers.

Leslie Vosshall

Below is a point-by-point response to reviewers' comments:

Reviewer #1 (Remarks to the Author):

This is an interesting study that describes for the first time the use of the Q-system of binary expression in the mosquito. It is used here to visualize ORNs in *Anopheles*. The study produces a map of antennal lobe glomeruli that should be very useful to the field. Overall this is an impressive study that makes an important advance in the field of vector biology.

1. The paper should cite the work of Sylvia Anton on the projections of ORNs in the antennal lobe of *Anopheles*.

We have now included the Anton et al. *Arthropod Structure & Development* (2003) and Ghaninia et al *Arthropod Structure & Development* (2007) citations in regards to ORN projection in *Anopheles*.

2. p. 7 Was there weak, un-induced expression of the reporter in the MB and optic lobe neurons in both of the QUAS-mCD8:GFP transgenic lines?

Yes, there was the same background expression found in both QUAS-mCD8:GFP transgenic lines. This has been clarified in the revised text.

3. p. 8 There are twice as many ORNs as Orco+ cells. The authors suggest that many sensilla contain no Orco+ cells. Another possibility is that many sensilla contain one Orco+ and one Orco- cell. Do the authors know from the labeling patterns whether most labeled sensilla contain one or two labeled cells?

Olfactory neurons densely occupy the antennae. This made it difficult to visualize which olfactory neuron cell bodies corresponded to which sensilla. As such it was difficult to determine if sensilla were targeted by 1 or 2 *Orco-QF2>QUAS-mCD8:GFP* neurons.

4. p. 9 GFP fluorescence was observed in the hindgut. Was there any in the male reproductive track? Zwiebel's group has reported expression of Ors in *Anopheles* sperm (PNAS, 2014).

As also detailed in the response to Reviewer 2 regarding hindgut expression, we examined many internal tissues of both male and female adult *Anopheles* mosquitoes in *QUAS-mCD8:GFP* control and *Orco-QF2, QUAS-mCD8:GFP* transgenic animals, by staining for anti-CD8, anti-DmOrco, and examining GFP and DAPI. What we found was intriguing, but requires extensive additional work to be conclusive. We found that hindgut cells were GFP+, and weakly Orco+ immunoreactive, but not in all samples. It could be that Orco expression is regulated based on the feeding status of the female, which we did not properly control. We also found what appears to be labelling in the testis, likely sperm as described previously (Pitts, PNAS, 2014), as well as an un-identified tissue that is part of the ovaries. In all these cases, anti-DmOrco staining was weak. While the investigation of non-olfactory internal tissues is indeed fascinating, we feel that they require careful and proper attention that is beyond the scope of a manuscript focused on olfactory tissues. We prefer to take the necessary time, and setup the necessary collaborations, in order to rigorously examine internal *Orco-QF2* expression. As such, we have removed mention of the hindgut expression from the revised manuscript, as well as refrained from commenting on potential sperm

labelling, and now instead say that the *Orco-QF2* analysis was focused only on external tissues in the adult.

5. p. 10 "clearly labeled commissure between the left and right ALs." It would be helpful to indicate these with an arrow in both males and females. Also, the male and female images in Fig. 3a appear very different and this difference needs to be explained to those who are not familiar with antennal lobes.

We have updated Figure 3 to more clearly show the commissures in the brain examples. We have also expanded on the description on the differences between the male and female antennal lobes in the revised manuscript. We 3D traced male and female brains, as well as male and female antennal lobes, and found that the female antennal lobes were ~1.89 times the volume of male antennal lobe. This cannot be accounted for based on the size of the brains- females brains were ~1.07 times the volume of male brains. This new data is provided in Supplemental Source Data 4.

6. p. 10 Antennal ORNs project only to the ipsilateral side. Maxillary projections are bilateral. --The work of Anton and others should be cited here.

The work of Anton and Ignell have now been cited in the revised manuscript.

--The *Anopheles* antenna seems to differ from the *Drosophila* antenna. The authors should discuss additional species so readers will know whether one of these two species is an outlier.

Along with literature searches and enlisting guidance from a neuro-entomologist formerly from the Hildebrand lab, we found that the only insects that have bilaterally projecting ORNs from the antennae are the Brachyceran flies (higher flies defined as having aristate antennae, including *Drosophila*). So differences in innervation patterns between *Anopheles gambiae* antennal neurons and *Drosophila melanogaster* antennal neurons reflects their membership in the order of Brachyceran (*Drosophila*) or Nematocera (*Anopheles*).

7. p. 11 in contrast to *Drosophila*, the appearance of the AL varies markedly among animals. This deserves comment and discussion; the reader will wonder whether this interspecific difference arises from biology (e.g. more genetic variability in the mosquito population) or technical considerations.

We have expanded on this point in the section 4.6 of Methods. We clarify that experimental conditions were maintained as much as possible, but differences between antennal lobe samples likely arose due to a combination of technical (dissections and mounting) and biological reasons (e.g., a rigid esophagus pressing against antennal lobes deforming glomeruli to different degrees). Future transgenic labelling of individual olfactory neurons will help address the question of glomerular variability.

8. p. 12 ... "this implies that the same olfactory receptors may be expressed in the animals of both genders...." The logic here is not clear. The results discussed in this paragraph would seem to be compatible with a model in which different Ors are expressed in males and females. (There is no citation in this para. for genomic analysis.) Also, it's not clear to the casual reader why the assumption of one Or/ORN needs to be invoked here.

We agree that the logic of this section was confusing, and our data could be interpreted in a number of equally possible ways. We have removed this text from the revised manuscript.

9. p. 14 "... the labellum....might also be used to identify short-range host volatiles released during a successful bite. If such volatiles are not detected, the mosquito continues probing." How often does Anopheles bite the same individual twice in rapid succession? A reference would be helpful here.

We have removed this section of the Discussion, as it pertained to discussion of a 'flavor' center that has also been removed.

10. Fig. 4A key: it's difficult to distinguish the darker blue from the lighter blue that is supposed to designate the "not labeled" glomeruli.

We have changed the colors in a revised Figure 4 to make it easier to see the labelled glomeruli, as well as are now consistent through the figure.

11. Figs. 3cd needs better explanation - especially because the backfill labeling is very dim. Perhaps arrowheads would help.

We have now included an outline of the backfilled glomeruli in Figure 3c, 3d, and 3e, which helps accentuate the dim backfill. We were hesitant to increase the gain of the backfill during post-processing, as this may inaccurately suggest to the reader that backfills can label glomeruli as robustly as the genetic labels.

Reviewer #2 (Remarks to the Author):

The work of Riabinina and colleagues is a straightforward development and analysis of transgenic tools in the mosquito *Anopheles gambiae* to visualize the subset of olfactory receptor neurons (ORNs) that express the OR coreceptor ORCO. The technical innovation in applying the QF bipartite expression system to *A. gambiae* is very nice (although not completely ground-breaking; for example, the Gal4/UAS system has already been established in this species, as acknowledged by the authors). Nevertheless, the exploitation of their tools to document the anatomical properties of ORNs helps confirm, correct and extended previous histological studies in this species, which relied entirely on the (imperfect) neuronal backfill method to relate peripheral chemosensory organs to central projection domains. Although the expected readership is probably relatively limited (mosquito and other insect olfactory researchers), it is nevertheless an important piece of work for bringing cutting-edge genetic tools into ecologically and biomedically-relevant species.

My main concern is the surprising observation that only 33 glomeruli are labeled with ORCO-QF, because this is inconsistent with previous expression data (from the Zwiebel lab) indicating that about twice as many ORs are expressed in the antenna. As the authors mention, this either implies a departure from the 1 OR:1 ORN rule, or that their defined glomeruli boundaries actually represent more than one glomeruli, but I find both of the explanations unlikely (at least as the necessary high frequency to make the numbers match). Also, the innervation of many trichoid sensilla with non-ORCO positive ORNs would be different from all other characterized insect species, which is possible but surprising. In addition, this unexpectedly low number of ORCO glomeruli also leaves roughly 40 "vacant" glomeruli, which the authors suggests would be innervated by IRNs. However, the Zwiebel lab data suggest that only about half this number of IRs are expressed in the antenna (i.e.,

not enough for occupying all these glomeruli). These discrepancies make me wonder whether the ORCO-QF line is as faithful as the authors argue based upon quantification of ORCO/CD8 co-staining. If it is true that antibody staining on whole mount is not entirely reliable, is it possible that they authors may have missed visualizing a large number of endogenous ORCO-positive cells that do not express the ORCO-QF. If I understand their quantifications, they see staining of, for example, 35-70 ORCO/CD8 positive cells per female antenna, but this only a very small fraction of the 500-600 GFP labeled cells (visualized without staining). (This contrasts with larvae, where it appears that there is an excellent correspondence between anti-ORCO, anti-CD8, and GFP+ cells). If the adult appendages are understained, then more thorough validation experiments are need (for example, by staining antennal sections, where antibody penetration is less likely to be problematic) because the faithfulness of the ORCO-QF line is key to many of the observations and interpretations.

We appreciate the concerns of the reviewer. We have performed the requested experiments, which now appear as new Supplemental Figure 7. We sectioned antenna from adult female *Orco-QF2*, *QUAS-mCD8GFP* mosquitoes, and labelled them with anti-DmOrCO. We examined the overlap between Orco antibody labelled cell bodies and sensilla to those that were GFP-labelled. We chose to mainly focus on sensilla in this analysis since both Orco protein and the mCD8:GFP marker localize to dendritic membranes. The colocalization in sections of cell bodies may not be as accurate as Orco localizes to vesicles that are being transported to the cell membrane surface, whereas mCD8:GFP localizes mainly to membrane surfaces. We found, as previously reported in Fig 2, that all Orco-labelled sensilla were also GFP+. In addition, we found that all GFP+ sensilla were also Orco+ (new Source Data 3). This suggests that the Orco-QF2>QUAS-mCD8:GFP labelling likely labels only Orco+ neurons, consistent with our original analyses. We have also expanded our discussion pertaining to the difference between the number of reported OR genes and the number of ORCO+ glomeruli.

We also should clarify a point of confusion from our quantifications. In data the reviewer refers to in which we label ~70 Orco/CD8 per female antennae, these were actually confocal images of antennal segments, not full antennae. The full antennae consistently were labelled by ~600 GFP+ cells. We apologize for this confusion, and have now made it clear in our supplied tables that we are quantifying antennal segments, not full antennae, in those data.

Minor comments:

- it would be interesting to confirm or refute the hindgut expression of the ORCO-QF line with antibody staining (rather than leaving this only as a provocative, but potentially artifactual, expression pattern).

We agree it is important to distinguish if this is real *Orco-QF2* expression, or if it represents background *QUAS-mCD8:GFP* expression. We performed the suggested experiments, and examined the gut of both male and female adult *Anopheles* mosquitoes in *QUAS-mCD8:GFP* control and *Orco-QF2*, *QUAS-mCD8:GFP* transgenic animals, by staining for anti-CD8, anti-DmOrco, and examining GFP and DAPI. What we found was intriguing, but requires extensive additional work to be conclusive. We found that hindgut cells were GFP+, and weakly Orco+ immunoreactive, but not in all samples. As part of the gut prep, additional

tissues were also included. We also found what appears to be labelling in the testis (likely sperm as described previously (Pitts, PNAS, 2014) as well as an un-identified tissue that is part of the ovaries. In all these cases, anti-DmOrco staining was weak. While the investigation of non-olfactory internal tissues is indeed very interesting, we feel that they require careful and proper attention that is beyond the scope of a manuscript focused on olfactory tissues, and could not be fully presented in a single supplemental figure. We prefer to take the necessary time, and setup the necessary collaborations, in order to rigorously examine internal *Orco-QF2* expression. As such, we have removed mention of the hindgut expression from the revised manuscript, and now instead say that the *Orco-QF2* analysis was focused only on external tissues in the adult.

- the authors makes observations counter to previous work from Zwiebel (regarding labellar projections to the antennal lobe) and Ignell (whose proposed mechanosensory center is actually composed of ORCO-positive neurons). Could they elaborate on the basis for the previous observations and why they observe different results?

We are uncertain why labellar projections in previous work were found to weakly innervate the antennal lobe. This type of innervation would have been obvious in GFP+ labellar neurons in our transgenic lines. We repeated the neurobiotin labella fills exactly as reported in the Zwiebel work (Kwon, 2006), and in >12 samples of labellar fills we did not observe innervation in the antennal lobe. We also tried alternative conjugated dyes and modified filling approaches, but in no instances found innervation in the antennal lobe.

Regarding the Johnston's organ mechanosensory center in the antennal lobe as formulated by Ignell: from what we can tell by a careful reading of the literature, this was based on assumptions from EM sections that nerve bundles innervating the antennal lobes originated from the Johnston's organ or from the antennal nerve. The authors did not present direct experimental evidence that the Johnston's organ innervates the antennal lobe. Such evidence will be very challenging to acquire using neurobiotin dye-filling as this requires severing the antennae before the Johnston's organ segment. Even this would mainly label antennal nerves, which might make it difficult to dis-entangle the source of antennal lobe innervations. As such, previously state-of-the-art techniques have not allowed distinction between antennal and Johnston's organ nerve bundles.

- the Introduction and Discussion could be shortened. There is quite laborious discussion of the background to quite standard bipartite gene control systems, and, in particular, the health impact of mosquito vector research, which seems unnecessary and/or irrelevant to the present manuscript. As one example, lines 323-337 could be removed without losing any impact on this manuscript; it's fine to mention the idea of a flavor center but without any functional data it remains speculative.

We have greatly shortened the Discussion as suggested by the reviewer. In particular, we have deleted the discussion pertaining to 'flavor' that included lines 323-337. We have also shortened discussion pertaining to the 3 *Orco*-,MP+ as suggested by Reviewer 3. We did not alter the introduction as dramatically as we felt that many target readers (those for example interested in vector control) may not be as familiar with binary expression approaches, and those working on *Drosophila* olfaction may not be as familiar with mosquito olfaction.

- figure 1a - are the scale bars correct? (notably the "100 mm" scale bars)

We apologize for this error that was caused by conversion between Mac and PC Illustrator files. It has been corrected in the Revised Figure 1.

- in the co-expression analyses, it would be helpful to show the merged as well as the separate channels.

We have now included the merged channels in revised figures Figure 2, Figure 3, and Supplementary Figure 6 and new Supplementary Figure 7.

Reviewer #3 (Remarks to the Author):

Riabinina et al. present a major technical breakthrough in neurogenetics by establishing the bipartite QF/QUAS transcriptional activation system in the malaria mosquito, *Anopheles gambiae*. This is the first time that the Q system has been shown to work outside of the conventional model organisms, and this paper opens up an entire field of neural circuit analysis in this important vector mosquito. Here the authors label all cells expressing the olfactory receptor co-receptor (*orco*) with a membrane-tethered GFP and examine the neuroanatomy of *orco* in larvae and adults. While most of what was discovered was expected based on earlier anatomical work in the mosquito, and older work in *Drosophila melanogaster*, there are a few surprises. Although female mosquitoes have more than twice as many *orco*-positive olfactory neurons than males, the number of antennal lobe glomeruli is comparable between the sexes. *Orco*-positive fibers innervate the subesophageal zone, which has not previously been shown to receive olfactory input. Overall the work is technically sound, and opens up a new field of neurogenetics in the mosquito. The work would benefit from some revisions of text and figures as detailed below.

MAJOR POINTS

1. The title and abstract emphasize the organization of olfactory centers, but the work is as much or more about the Q system working in the mosquito. To my taste the anatomical findings about the organization of sensory centers in the adult brain are solid, but down the road this paper will be read and cited because it is the first use of QF/QUAS in mosquitoes. That should be made clearer in the title and abstract.

We discussed this point at length. We concluded that we prefer to emphasize the biology of the findings, even if that was at the expense of emphasizing the introduction of the Q-system into *Anopheles*. The introduction of new genetic tools into *Anopheles* is fairly rare, and so this would still be appreciated by the community. Due to journal restrictions on abstract length, we could not expand upon this point in the abstract. We modified the abstract sentence "Here, we introduced and employed the Q-system" to highlight that this is the first use of the Q-system in *Anopheles*.

2. At three places in the paper, the authors provide extensive speculation that I feel goes too far beyond the data to support it.

The first is the assertion that most projections are ipsilateral, but a subset are contralateral. The ablation experiments are suggestive, but not definitive, as the kind of clonal analysis done routinely in *Drosophila* would be. The images provided are not helpful for the reader to judge the evidence for contralateral projections. I suggest backing off a bit from the claims, and put in images that show the commissure.

We have provided higher magnification images that show more clearly the commissures, as requested also by Reviewer 1. From our dye filling of antennae and maxillary palps, as well as similar from other groups (Anton, Ignell, and others), we are confident that *Anopheles* antennal nerves project only ipsilaterally, and that the maxillary palp neurons project bilaterally. Note, *Aedes aegypti* maxillary palp olfactory neurons may only innervate ipsilateral antennal lobes (Ignell, J Comp Neurol 2005, p.218).

The second point is the assumption that the "missing" glomeruli in the antennal lobe must express gustatory receptors (GRs) or ionotropic receptors (IRs). The authors present this as a foregone conclusion, and while it is certainly likely, the mosquito is already showing us some surprising results that differ dramatically from the situation in the fly. So I would not go so far as to call the circuit solved based on assumptions that the mosquito and fly antennal lobe will be homologous.

We agree with the reviewer that we have only indirect evidence that the Orco-negative glomeruli are innervated by GRNs or IRNs. We have thus modified the text to reflect this ambiguity.

The third point is the evolutionary arguments that compare the mosquito and fly antennal lobe, specifically the numerology of the presumptive carbon dioxide sensitive glomeruli. The authors interpret their staining to be 3 separate glomeruli, and suggest that this might receive different types of activity. There is not enough evidence in the pictures alone to conclude this. In fact, there is not evidence presented that these are indeed the carbon dioxide glomeruli. Again, a more nuanced discussion that allows some uncertainty is recommended. It would be nice to go beyond just comparing the anatomy between mosquito and fly, perhaps to the moth, honeybee, and ant glomeruli for which there are 3-D atlases.

We agree that it might be premature to claim that all three Orco-, MP+ glomeruli would be innervated by the CO₂-sensing GRNs. We have deleted the extensive discussion pertaining to this from the revised manuscript, as well as introduced discussion as to what other sensory neurons might project to these glomeruli from the maxillary palp.

Regarding a comparison of 3-D antennal lobe atlases. This would be very interesting, but without knowledge of which glomeruli are innervated by ORCO+, IR+, or GR+ neurons in these other insect atlases, the comparison would not be as informative.

3. A few interesting puzzles emerge from the results, that I would like to see the authors comment on. The first is the observation that female mosquitoes have dramatically more sensory neurons but a similar number of antennal lobe glomeruli. Are the male glomeruli much smaller than the female? If not, do the authors have any thoughts about how the male brain is compensating for much reduced input?

We have now quantified the volumes of male and female *Anopheles* brains (animals reared and then imaged under identical conditions), as well as the volumes of the male and female antennal lobe glomeruli. This analysis now appears in the revised manuscript in section 2.4. Female brains ($5121160 \mu\text{m}^3 \pm 274266 \mu\text{m}^3$, mean \pm sem, n=3) were ~7% larger in volume than male brains ($4783181 \mu\text{m}^3 \pm 297216 \mu\text{m}^3$, n=3). In contrast, female antennal lobes ($165765 \mu\text{m}^3 \pm 12987 \mu\text{m}^3$, n=5) were 1.89 times larger than male antennal lobes ($87599 \mu\text{m}^3 \pm 2181 \mu\text{m}^3$, n=3) reflecting increases in individual glomerular volumes (new Source Data 4). This is reflected in the antennal lobe images in Figure 3a, in which the male antennal lobes are smaller than female antennal lobes (note the scale bars are the same in

both images). The increase in antennal lobe size in females is in agreement with a larger number of Orco+ receptor neurons in the antennae and maxillary palps innervating antennal lobe glomeruli.

The second is why are there only 33 orco glomeruli and 79 ORs? A discussion of this mismatch would be interesting.

As mentioned in the response to the editor, answering this question undoubtedly requires follow-up neurogenetic experiments. Nonetheless, we did try to partially address this question in the revised manuscript. For one, of the 79 AgORs, only 59 are expressed in adult antennae and maxillary palp at transcript levels 2-fold more in ant/mp vs body as determined by RNA-seq (Pitts, 2011). Furthermore, there is precedence that many AgORs maybe co-expressed in olfactory neurons. For example, there are genomic clusters of AgORs that appear to be co-expressed (e.g., cluster 1B:OR13, OR15, OR16, OR17, OR47, OR55; Schultze, 2013; Karner, 2015). Examining the *Anopheles gambiae* genome (VectorBase, *AgamP4*), there are 6 such genomic clusters in which ORs are within 500 bps of each other. Although these remain to be experimentally validated, it suggests that 17 ORs maybe expressed in as few as 6 different ORN types. Similarly, in *Drosophila*, OR gene clusters can be co-expressed in the same neuron (eg., Or22a/Or22b; Or65b/Or65c), yet other ORs can be co-expressed in the same neuron outside of gene clusters (e.g., Or19a/Or19b; Or33c/Or86e). In sum, the number of distinct ORCO+ olfactory neurons targeting the antennal lobe may be closer to the number of Orco+ glomeruli that we have identified. We have included this discussion in the revised manuscript.

4. The authors introduce the concept that the mutual innervation of orco neurons of the antennal lobe and the SEZ is evidence for smell/taste fusion, or the emergence of "flavor." This is an important and interesting concept, but my sense is that not all the data are in yet to draw this strong psychophysical conclusion.

We agree with the reviewer that a 'flavor' center was too speculative. We have revised this to mention that this might be a brain region of sensory integration. We have also deleted the section about the 'flavor' center from the discussion.

5. Because larval expression is not integral to the paper, Figure 1a and all discussion of larval expression could be removed from the manuscript. A future analysis of the larval chemosensory system is a better place to feature such data.

We discussed this point at length. We concluded that the larval data is strong support for the fidelity of the *Orco-QF2* expression pattern, including the quantification of Orco+ and GFP+ overlap, and as such, strengthened the claims of the manuscript in regards to the adult *Orco-QF2* expression pattern. We imagined that removing the larval data and only mentioning it as data not shown might cast doubts on the genetic labelling.

Nonetheless, we did agree that the larval data did make the main figures more difficult to follow as a whole, especially in light of an emphasis on adult expression patterns in Figure 3 and Figure 4. As such, we moved the larval data from Figure 1 and Figure 2 into new Supplemental Figures (Supplemental Figure 3 and Supplemental Figure 6).

FIGURE SUGGESTIONS

Figure 1 Given the bleed-through of 3xp3 and the scattered constitutive expression of QUAS-CD8-GFP heterozygote with no driver, QUAS-CD8-EGFP and not wild-type should be the control in these staining experiments.

We have replaced the wild-type images with QUAS-mCD8:GFP animals in the revised Figure 1 and Supplementary Figure 3.

Figure 1: Beyond quantifying the number of ORCO+ cells in each antenna, the authors could take this one step further and quantify the number of ORCO+ cells per segment. This might reveal something about the uniformity of ORCO expression, and suggest the composition of each segment as an identical repeating unit or not.

The quantification of ORCO+ cells per segment now appears in Source Data 1.

Figure 2, and Figure 3c-e: in addition to the red, green, and blue panels, please provide merged views of all samples so that it is possible to compare the staining in a single image.

We have included the merged views in the revised Figure 2 and Figure 3c-e.

Figure 4a, b: use the same color scheme for marking the glomeruli according to orco gene expression in both panels a and b. Currently it is quite confusing to go from one panel to the next.

We have revised the color scheme as requested. The colors used in the revised Figure 4a are now consistent to the colors used in Figure 4b.

Figure 4: A major difference in this work relative to previously published work is that the JOC is not mechanosensory but olfactory. If the authors have any insights in where it projects, perhaps by dye-filling the JOC, that would be helpful. Does it go to AMMC or does it also send projections to these ORCO+ glomeruli?

We only performed neurobiotin labelling of severed antennae, and so could not visualize Johnston's organ nerves. To label Johnston's Organ nerves with neurobiotin fills will be very challenging to acquire, as this requires severing the antennae before Johnston's organ. This will still predominately label antennal nerves, making it difficult to dis-entangle the source of antennal lobe innervations. The identification of Johnston's organ brain targets will likely require future genetic labelling.

Figure 4c: the green shading is not visible on the actual images, please check or select another shading color.

We have changed this to an easier to see orange shading in the revised Figure 4c.

Minor points:

1. L36 remove "SOG" as an abbreviation, it has been superseded by "SEZ" per the Kei Ito neural nomenclature system.

"SOG" appears only in the "Keywords" section, and nowhere else in the manuscript. It was purposefully included in "Keywords" to capture those who might not be aware of the nomenclature change to SEZ.

2. L189 word choice, gender should probably be "sex"

Corrected, thank you.

3. L214 I suggest that the statement about "high background" of the anti-orco antibody be put into context. The antibody (generated by my lab) is selective for orco in *Drosophila melanogaster* and *Aedes aegypti*, as verified by the absence of staining in the null mutants generated in each species. Without an orco null mutant in *Anopheles gambiae*, it is hard to state for certain that the staining is background or signal. A more cautious disclaimer is needed here.

We have corrected this to qualify that the background is present only in *Anopheles* tissues.

4. L257 typo positive

Corrected, thank you.

5. L350-353 other groups will be interested in whether any other transgenes were attempted that did not give expression.

We have now included in the Methods section that we also attempted to generate *GR24-QF2*, *elav-QF2* and *nsyb-QF2* putative promoter transgenic lines, but did not get expression from these lines.

6. The Figure legend for Figure 1b refers to larvae, whereas I think the images are from adult antenna.

Figure 1b were indeed larvae. The larvae figures in Figure 1 have been moved to Supplemental Figure 3, so that the main figures now only represent adult.

7. State in the manuscript that the antennal lobe atlas will be available for others to download and work with.

We had cited the availability of the atlas as Supplemental Models, but have now made this more explicit in the revised Methods.

Reviewers' Comments:

Reviewer #1 (Remarks to the Author)

My comments have all been addressed in a satisfactory way.

Reviewer #3 (Remarks to the Author)

The authors have done an excellent job responding to my concerns, and those of the additional reviewers. The rather heroic antennal sectioning experiments are extremely convincing on the point that the QF system is faithfully representing endogenous orco expression. This was a major concern of Reviewer 2, and I completely agree it was important to show it. And the authors have shown it brilliantly.

The precise volume measurements of male/female antennal lobes further solidify the observed sexual dimorphism that lacked quantification in the original submission.

The larval data are much more effective in the Supplement and I am happy to see them there.

Finally, the authors have cleaned up the writing, made it less speculative, and have been more generous in citing previous work in other insects.